# Consistent Point Data Assimilation in Firedrake and Icepack

Reuben W. Nixon-Hill[1,2], Daniel Shapero[3], Colin J. Cotter[1], and David A. Ham[1]

[1]Department of Mathematics, Imperial College London, London, SW7 2AZ
[2]Science and Solutions for a Changing Planet DTP, Grantham Institute for Climate Change and the Environment, Imperial College London, London, SW7 2AZ
[3]Polar Science Center, Applied Physics Laboratory, University of Washington, 1013 NE

**Correspondence:** Reuben W. Nixon-Hill (now at UK Met Office, reuben.nixonhill@metoffice.gov.uk, https://orcid.org/0000-0001-6226-4640)

**Abstract.**

We present a high-level, differentiable and composable abstraction for the point evaluation of the solution fields of partial differential equation models. The new functionality, embedded in the Firedrake automated finite element system, enables modellers to easily assimilate point data into their models at the point locations, rather than resorting to extrapolation to a computational mesh. We demonstrate the expressiveness and ease with which more mathematically defensible data assimilation can be performed with examples in the fields of groundwater hydrology and glaciology.

In many disciplines of the geosciences, modellers wish to estimate fields that are difficult to observe directly from measurements of fields that are. These measurements are often sparse and it is common practice to first extrapolate these measurements to the grid or mesh used for computations. When this estimation procedure is viewed as a deterministic inverse problem, the extrapolation step is undesirable because the choice of extrapolation method introduces an arbitrary algorithmic degree-of-freedom that can alter the outcomes. When the estimation procedure is instead viewed through the lens of statistical inference, the extrapolation step is undesirable for the additional reason that it obscures the number of statistically independent measurements that are assimilated and thus makes it impossible to apply statistical goodness-of-fit tests or model selection criteria. The introduction of point evaluation into Firedrake, together with its integration into the automatic differentiation features of the system, greatly facilitates the direct assimilation of point data and thus improved methodology for solving both deterministic and statistical inverse problems.

## 1 Introduction

Many disciplines in the earth sciences face a common problem in the lack of observability of important fields and quantities. In groundwater hydrology, the conductivity of an aquifer is not directly measurable at large scales; in seismology, the density of the earth; and in glaciology, the viscosity of ice. Nevertheless, these are necessary input variables to the mathematical models that are used to make predictions. *Inverse problems* or *data assimilation* have us estimating these immeasurables through some mathematical model that relates them to something measurable. For example, the large scale density (an immeasurable) and displacement (a measurable) of the earth are related through the seismic wave equation. Combining measurements of

displacement or wave travel time from active or passive seismic sources can then give clues to the density structure of the earth. Estimating an unobservable field is often posed as an optimisation problem with a partial differential equation (PDE) as a constraint. The objective consists of a model-data misfit metric[1] and a regularisation term to make the problem well-posed. This estimation procedure can be viewed either as a deterministic inverse problem, or through the lens of Bayesian inference, where it amounts to finding the maximizer of the posterior distribution.

Geoscientists use a wide variety of measurement techniques to study the Earth system. Each of these techniques yields data of different density in space and time. The number of independent measurements and their accuracy dictates how much information can be obtained about unobservable fields through data assimilation. The tools used to solve these PDE-constrained optimisation problems do not always allow the varied density in space and time of such measurements to be taken into account. In such scenarios it is common practice to use a model-data misfit which treats such measurements as a continuous field. Exactly how to create such a field from discrete measurements is a choice the modeller must make. Alternatively, if a tool allows it, one can create a model-data misfit that is evaluated at the discrete points. This distinction has important consequences. For example, if observations are sparse then using a misfit which treats measurements as a continuous field requires significant assumptions.

The models we are interested in are discretisations of differential equations to which we find some numerical solution. Galerkin methods, here referred to generally as finite element methods, are a popular approach with useful properties. Key among these properties is the high level of mathematical abstraction with which finite element models can be expressed. This has enabled the creation of very high level finite element tools such as Firedrake (Ham et al., 2023; Rathgeber et al., 2016) and FEniCS (Logg, 2009; Alnæs et al., 2015) in which the user writes a symbolic mathematical expressions of the PDEs to be solved in the Unified Form Language (UFL) (Alnæs, 2012) domain specific language. Parallelisable, scalable (Betteridge et al., 2021) and efficient finite element C code implementing the model is generated, compiled, and executed automatically. Here we will employ Firedrake. The symbolic representation of the PDE model also enables gradients and Hessian actions to be automatically computed using the discrete adjoint generation system dolfin-adjoint/pyadjoint (Mitusch et al., 2019). Firedrake supports a wide array of elements and has been used to build the ocean model Thetis (Kärnä et al., 2018), atmospheric dynamical core Gusto (Ham et al., 2017), glacier flow modelling toolkit Icepack (Shapero et al., 2021), and the geodynamics system G-ADOPT (Davies et al., 2022).

Firedrake already includes the ability to evaluate fields at discrete point sets. To use model-data misfit functionals that are evaluated at discrete points in PDE-constrained optimisation problems, we need to be able to compute a first or second derivative of the point evaluation operation using reverse-mode automatic differentiation. Here we will show how point cloud data can be represented as a finite element field and hence how point evaluation can be incorporated with the automatic code generation and automatic differentiation capabilities of Firedrake and dolfin-adjoint/pyadjoint.

We will use these new capabilities to compare the two model-data misfit approaches. We go on to demonstrate its use for influencing experiment design in groundwater hydrology, where measurements are generally very sparse. Lastly we perform

---

[1]In weather and climate models, this is referred to as *variational* data assimilation.

a cross-validation data assimilation experiment in glaciology using Icepack (Shapero et al., 2021). This experiment requires model-data misfit terms that use point evaluations and allows us to infer information about the statistics of our assimilated data.

We show how we can integrate point data into the finite element method paradigm of fields on meshes and demonstrate that this can be used to automate point evaluation such that we can automatically solve these minimisation problems.

Point data assimilation is, of course, not new. In the field of finite element models, it has been hand coded into models on numerous occasions. For a previous example in the Firedrake framework, see Roberts et al. (2022). Automated systems from a higher level user interface are less common, but an important example is hIPPYlib (Villa et al., 2021), which provides an inverse problem layer on top of FEniCS. The distinction between that work and this is that HIPPYlib implements point evaluation as a layer on top of UFL, while this work extends UFL to support point data intrinsically. The advantage of this approach is that it is fully composable: the output of a point evaluation operation can be fed directly back into another operator that expects a UFL input.

The paper is laid out as follows. Sections 2, 3 and 4 describe how we integrate point data with finite element methods while Sect. 5 shows our specific Firedrake implementation. In Sect. 6 we return to the topic of data assimilation and pose the question of model-data misfit choice in more detail; we investigate this in Sect. 7.1 using our new Firedrake implementation. The further demonstrations in groundwater hydrology and glaciology can be found in Sections 7.2 and 7.3, respectively.

## 2   Finite element fields

In finite element methods the domain of interest is approximated by a set of discrete cells known as a mesh $\Omega$. The solution of a PDE $u$ is then approximated as the weighted sum of a discrete number of *basis* or *shape* functions on the mesh. For $N$ weight coefficients and basis functions our approximate solution

$$u(x) = \sum_{i=0}^{N-1} w_i \phi_i(x) \tag{1}$$

is called a *finite element field*. The set of basis functions $\{\phi_i(x)\}$ are kept the same for a given mesh but the weights $\{w_i\}$ are allowed to vary to form our given $u(x)$: these weights are referred to as Degrees of Freedom (DoFs).

The set of all possible weight coefficients applied to the basis functions on the mesh is called a finite element function space $\mathrm{FS}(\Omega)$, here referred to as a *finite element space*. A finite element field is therefore a member of a finite element space

$$u \in \mathrm{FS}(\Omega). \tag{2}$$

The approximate weak solutions to PDEs produced by the finite element mentod are finite element fields. As long as this field is continuous we know its values unambiguously. We can therefore evaluate the solution at the location of any measurement so long as that location is within the boundaries of the mesh. We will return to finite element spaces with discontinuities later.

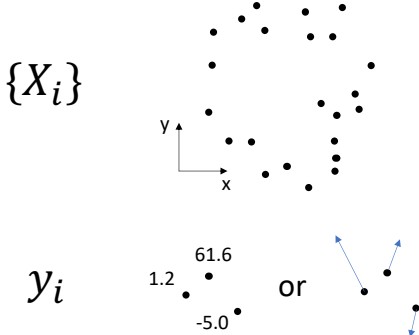

**Figure 1.** Point data consist of (a) a set of $N$ spatial coordinates $\{X_i\}_{i=0}^{N-1}$ (a 'point cloud') and (b) the scalar, vector or (not shown) tensor values $\{y_i(X_i)\}_{i=0}^{N-1}$ at those coordinates. Maintaining this distinction is key when trying to form a rigorous way of handling point data for any numerical method, such as the finite element method, which separates the idea of domain and values on the domain.

## 3 Point data

In order to seamlessly integrate point data as a first class object in the finite element paradigm, it is necessary to look carefully at what is meant by point data. Point data can be separated into two parts: (a) the locations $\{X_i\}$ of the $N$ data points at a given time (a point cloud) and (b) the $N$ values $\{y_i(X_i)\}_{i=0}^{N-1}$ associated with the point cloud (see Fig. 1). Finite element fields have a similar distinction: (a) the discretised shape of the domain of interest (the mesh $\Omega$) and (b) the values associated with that mesh (the weights applied to the basis functions).

Applying the finite element distinction to the locations of the data (a) suggests a 'point cloud mesh' formed of $N$ disconnected vertices at each location $X_i$:

$$\Omega_v = \{X_i\}_{i=0}^{N-1}. \tag{3}$$

We refer to this as a 'vertex-only mesh'. This is an unusual mesh: a vertex has no extent (it is topologically zero dimensional) but exists at each location $X_i$ in a space of geometric dimension $\dim(X_i)$. Fortunately meshes with topological dimension less that their geometric dimension are not unusual: 2D meshes of the surface of a sphere in 3D space are commonly used to represent the surface of the earth. Such domains are typically called 'immersed manifolds'. Disconnected meshes are also not unheard of: the software responsible merely needs to be able to iterate over all the cells of the mesh. In this case each cell is a vertex $X_i$. We can therefore legitimately construct such a mesh.

We now need to consider the values $\{y_i(X_i)\}_{i=0}^{N-1}$ (b). Only one value, be it scalar, vector or tensor, can be given to each cell (i.e. each point or vertex). Fortunately a finite element space for this case exists: the space of zero order discontinuous Lagrange polynomials

$$y \in \text{P0DG}(\Omega_v) \tag{4}$$

where P0DG stands for Polynomial degree 0 Discontinuous Galerkin. Here $y$ is a single discontinuous field that contains all of the point data values at all of the point locations (mesh vertices)

$$y(x) = \begin{cases} y(X_i) & \text{if } x = X_i, \\ \text{undefined} & \text{elsewhere.} \end{cases} \tag{5}$$

Integrating this over the vertex-only mesh $\Omega_v$ with respect to the zero-dimensional point measure $\mathrm{d}x_v$ gives

$$\int_{\Omega_v} y(x)\mathrm{d}x_v = \sum_{i=0}^{N-1} y(X_i) \quad \forall y \in \text{P0DG}(\Omega_v) \tag{6}$$

This definition enables direct and automated reasoning about point data in finite element language and yields useful results. So long as the locations of the vertices $X_i$ of a vertex-only mesh $\Omega_v$ are within the domain of its 'parent' mesh

$$\Omega_v \subset \Omega \tag{7}$$

then it is possible to map from some field $u$ in some finite element space defined on the parent mesh

$$u \in \text{FS}(\Omega) \tag{8}$$

to one defined on the vertex-only mesh

$$u_v \in \text{P0DG}(\Omega_v) \tag{9}$$

by performing point evaluations at each vertex location $u(X_i) \ \forall \ i$.

The operator for this can be formulated as finite element interpolation (known as 'dual functional evaluation' or simply 'dual evaluation') into P0DG, i.e.

$$\mathcal{I}_{\text{P0DG}(\Omega_v)}(;u) : \text{FS}(\Omega) \rightarrow \text{P0DG}(\Omega_v) \tag{10}$$

such that

$$\mathcal{I}_{\text{P0DG}(\Omega_v)}(;u) = u_v. \tag{11}$$

This operator is linear in $u$ which we denote by a semicolon before the argument. The construction of this operator is described in the next section.

## 4 Point evaluation as an interpolation operator

Most implementations of finite element methods distinguish between a set of 'global' coordinates covering the meshed domain and a set of 'local' coordinates defined on some reference cell. Computations on each mesh cell are implemented by transforming from global to local coordinates, performing an operation, then transforming the result back.

This section describes the process of evaluating parent mesh fields on the point cloud. We broadly follow the process in (Brenner and Scott, 2008, s.3.3). For a reference cell $\mathcal{K}$, local interpolation $\mathcal{I}_{\mathcal{P}(\mathcal{K})}$ for a set of $k$ local basis functions $\mathcal{P} = \text{span}(\{\tilde{\phi}_i\}_0^{k-1})$ of some locally defined field $\tilde{f}$ is given by the linear operator

$$\left[\mathcal{I}_{\mathcal{P}(\mathcal{K})}(;\tilde{f})\right](\tilde{x}) = \sum_{j=0}^{k-1} \tilde{\phi}_j'(;\tilde{f})\tilde{\phi}_j(\tilde{x}) \tag{12}$$

where $\{\tilde{\phi}_i'\}_0^{k-1}$ are the local dual basis linear functionals (the span of which are the nodes $\mathcal{N}$) and $\tilde{x}$ are the local coordinates. This uses Ciarlet's triple formulation $(\mathcal{K}, \mathcal{P}, \mathcal{N})$ (Ciarlet, 2002) as the definition of a finite element. Global interpolation over the entire mesh $\Omega$ for the complete finite element space $\text{FS}(\Omega)$, which we denote $\mathcal{I}_{\text{FS}(\Omega)}$, is the local application (i.e. transformed to local reference coordinates) of $\mathcal{I}_{\mathcal{P}(\mathcal{K})}$ to the globally defined field $f \in \text{FS}(\Omega)$.

Dual basis functionals are strictly a mapping from members of $\mathcal{P}$ to a scalar: i.e. the global interpolation operator for a given finite element space $\text{FS}(\Omega)$ is defined in all cases for fields in that space ($\mathcal{I}_{\text{FS}(\Omega)} : \text{FS}(\Omega) \to \text{FS}(\Omega)$). It is not unusual, however, for finite element libraries to allow interpolation from fields which are defined in another finite element space so long as the geometric dimension and meshed domains are consistent[2]. Both Firedrake and FEniCS allow this. This is a well-defined operation with understood approximation properties so long as the particular dual basis functionals are well-defined on the finite element space from which values are being interpolated (for more see Maddison and Farrell (2012)).

For $u_v = \mathcal{I}_{\text{P0DG}(\Omega_v)}(;u)$ where $u(x) \in \text{FS}(\Omega)$ we require, at each vertex cell $X_i$ of the vertex-only mesh $\Omega_v$, the point evaluation $u(X_i)$. The implementation of the global interpolation operator comprises the following for each vertex cell $X_i$ in our vertex-only mesh $\Omega_v$:

1. finding the cell of the parent mesh $\Omega$ that $X_i$ resides in,

2. finding the equivalent reference coordinate $\tilde{X}_i$ in that cell,

3. transforming the input field $u$ to reference coordinates on that cell giving $\tilde{u}$,

4. performing the point evaluation[3] $\tilde{u}(\tilde{X}_i)$ and

5. transforming the result back to global coordinates giving $u(X_i)$.

This operation formalises the process of point evaluation with everything remaining a finite element field defined on a mesh. These fields can have concrete values or be symbolic unknowns. If the symbolic unknown is a point, we can now express that in the language of finite elements. Whilst it is not the topic of this paper, we can now, for example, express point forcing expressions as

$$\int_{\Omega_v} \mathcal{I}_{\text{P0DG}(\Omega_v)}(;f(x))\,\mathrm{d}x_v = \sum_{i=0}^{N-1} f(x_i) = \sum_{i=0}^{N-1} \int_{\Omega} f(x)\delta(x - x_i)\,\mathrm{d}x_v. \tag{13}$$

---

[2]One usually also needs the finite element spaces to be continuous at cell boundaries for the operator to be well defined everywhere but this is not always explicitly checked for.

[3]For vector or tensor valued function spaces, this becomes the inner product of the point evaluation with a Cartesian basis vector or tensor respectively.

Given later discussion, note here that the 'interpolation' operation is exact and, excepting finite element spaces with discontinuities, unique: we obtain the value of $u$ at the points $\{X_i\}$ on the new mesh $\Omega_v$. A concrete example is considered in Sect. 7.

## 5 Firedrake implementation

**Listing 1.** An example point evaluation in Firedrake: the last three lines are the new functionality. In Firedrake, as in much finite element literature, fields and finite element spaces are known as functions and function spaces respectively.

```
 1: from firedrake import *
 2:
 3: def poisson_point_eval(coords):
 4:     """Solve Poisson's equation on a unit square for a random forcing term
 5:     with Firedrake and evaluate at a user-specified set of point coordinates.
 6:
 7:     Parameters
 8:     ----------
 9:     coords: numpy.ndarray
10:         A point coordinates array of shape (N, 2) to evaluate the solution at.
11:
12:     Returns
13:     -------
14:     firedrake.function.Function
15:         A field containing the point evaluatations.
16:     """
17:     omega = UnitSquareMesh(20, 20)
18:     P2CG = FunctionSpace(omega, family="CG", degree=2)
19:     u = Function(P2CG)
20:     v = TestFunction(P2CG)
21:
22:     # Random forcing Function with values in [1, 2].
23:     f = RandomGenerator(PCG64(seed=0)).beta(P2CG, 1.0, 2.0)
24:
25:     F = (inner(grad(u), grad(v)) - f * v) * dx
26:     bc = DirichletBC(P2CG, 0, "on_boundary")
27:     solve(F == 0, u, bc)
28:
29:     omega_v = VertexOnlyMesh(omega, coords)
```

```
30:        P0DG = FunctionSpace(omega_v, "DG", 0)
31:        return interpolate(u, P0DG)
```

Firedrake code for specifying and solving PDEs closely reflects the equivalent mathematical expressions. As an example see Listing 1, in which we solve Poisson's equation $-\nabla^2 u = f$ under strong (Dirichlet) boundary conditions $u = 0$ on the domain boundary, in just 8 lines of code. We specify the domain $\Omega$ and finite element space (here called a function space) to find a field solution $u \in \text{P2CG}(\Omega)$ (here called a function). Since Firedrake employs the finite element method we require a weak formulation of Poisson's equation

$$\int_\Omega \nabla u \cdot \nabla v - fv \, \mathrm{d}x = 0 \ \ \forall v \in \text{P2CG}(\Omega) \tag{14}$$

where $v$ is called a test function[4]. The specification of the PDE, written in UFL on line 25, is identical to the weak formulation. We apply our boundary conditions and then solve. Firedrake uses PETSc (Balay et al., 2022b, 1997) to solve both linear and nonlinear PDEs[5].

The solution is next evaluated at the points the user provides. The final three lines of Listing 1 follow the mathematics of Sect. 3 and Sect. 4. A vertex-only mesh $\Omega_v$ is immersed in the parent mesh $\Omega$ (Eq. 7) using a new `VertexOnlyMesh` constructor. The finite element space $\text{P0DG}(\Omega_v)$ is then created (Eq. 9) using existing Firedrake syntax. Lastly the cross mesh interpolation operation (Eq. 10) is performed; this again uses existing Firedrake syntax.

The vertex-only mesh implementation uses the DMSWARM point-cloud data structure in PETSc. The P0DG finite element space ins constructed on top of `VertexOnlyMesh` in the same way as a finte element space on a more conventional mesh.[6] The interpolation operation follows the steps laid out in Sect. 4.

The implementation works seamlessly with Firedrake's MPI parallelism. Firedrake performs mesh domain decomposition when run in parallel: vertex-only mesh points decompose across ranks as necessary. Where points exist on mesh partition boundaries a voting algorithm ensures that only one MPI rank is assigned the point. The implementation supports both very sparse and very dense points as we will demonstrate in Sect. 7.

Where solutions on parent meshes have discontinuities we allow point evaluation wherever it is well defined. Solutions on parent meshes from discontinuous Galerkin finite element spaces, which are not well defined on the cell boundaries, can still be point evaluated here: the implementation picks which cell a boundary point resides in.

All interpolation operations in Firedrake can be differentiated using a Firedrake extension to the dolfin-adjoint/pyadjoint package which allows us to solve PDE-constrained optimisation problems. This was a key motivating factor for turning point evaluation into an interpolation operation, and one which will be repeatedly used later.

---

[4]For those unfamiliar with the finite element method, this is standard nomenclature.

[5]Our solver functions are more flexible than this simple example suggests. For example, PETSc solver options can be passed from Firedrake to PETSc. See the PETSc manual (Balay et al., 2022a) and Firedrake project website https://www.firedrakeproject.org/ for more information

[6]Other mesh types in Firedrake use the DMPlex data structure rather than DMSWARM as explained in Lange et al. (2016). The construction of a finite element space (a function space), proceeds exactly as shown in Fig. 2 of Lange et al. but with DMPlex now a DMSWARM.

## 6  Assimilating point data

We start with some model, find $u \in \text{P2CG}(\Omega)$ such that

$$F(u, m) = 0 \tag{15}$$

where $m$ is a set of parameters; $u$ is the solution; and $F$ the equation, such as a PDE, that relates $m$ and $u$. Assume we have a set of evaluation points $\{X_i\}$ and data to assimilate at each point, $i$, given by:

$$u_{\text{obs}}^i \text{ at } X_i. \tag{16}$$

The objective is to solve the inverse problem of finding the parameters $m$ which yield these data.

A typical formulation involves running the model for some $m$ and employing a model-data misfit metric to see how closely the output $u$ matches the data $\{u_{\text{obs}}^i\}$. We then minimise the model-data misfit metric: the parameters $m$ are updated, using a method such as gradient descent, and the model is run again. This is repeated until some optimum has been reached.

The model-data misfit metric is a functional (often called the 'objective function') $J$ with a typical form being

$$J = J_{\text{misfit}} + J_{\text{regularisation}}. \tag{17}$$

The regularisation can be thought of as encoding prior information about the function space in which the solution lies. Often this uses known properties of the physics of the model (such as some smoothness requirement) and, in general, ensures that the problem is well posed given limited, typically noisy, measurements of the true field $u$.

A key question to ask here is "what metric should be used for the model-data misfit?" One approach, taken for example by Shapero et al. (2016), is to perform a field reconstruction: we extrapolate from our set of observations to get an approximation of the continuous field we aimed to measure. This reconstructed field $u_{\text{interpolated}}$ is then compared with the solution field $u$

$$J_{\text{misfit}}^{\text{field}} = \|u_{\text{interpolated}} - u\|_N \tag{18}$$

where $\|\cdot\|_N$ is some norm. We call the extrapolated reconstruction $u_{\text{interpolated}}$ since, typically, this relies on some 'interpolation' regime found in a library such as SciPy (Virtanen et al., 2020) to find the values between measurements. As we will see when we return to these methods in Sect. 7.1, $J_{\text{misfit}}^{\text{field}}$ is not unique since there is no unique $u_{\text{interpolated}}$ field. The method used to create $u_{\text{interpolated}}$ is up to the modeller and is not always reported.

An alternative metric is to compare the point evaluations of the solution field $u(X_i)$ with the data $u_{\text{obs}}^i$

$$J_{\text{misfit}}^{\text{point}} = \|u_{\text{obs}}^i - u(X_i)\|_N \ \forall\ i. \tag{19}$$

Importantly, $J_{\text{misfit}}^{\text{point}}$ is, with the previously noted discontinous Galerkin exception, unique and independent of any assumptions made by the modeller.

It is the difference between minimising $J_{\text{misfit}}^{\text{field}}$ and $J_{\text{misfit}}^{\text{point}}$ which we investigate here. Previously we could generate code to minimise a functional containing $J_{\text{misfit}}^{\text{field}}$ using Firedrake and dolfin-adjoint/pyadjoint. Dolfin-adjoint/pyadjoint performs tangent-

linear and adjoint mode Automatic Differentiation (AD) on Firedrake operations[7], including finding the solutions to PDEs[8] and performing interpolation. Point evaluation operations have been a notable exception. Now that Firedrake includes the ability to differentiate point evaluation operations by recasting them as interpolations, we can investigate minimising a functional which

contains $J_{\text{misfit}}^{\text{point}}$. As Listing 2 in Sect. 7.1 shows, this requires just a few lines of code. To the author's knowledge, the technology needed to minimise $J_{\text{misfit}}^{\text{point}}$ using automated code-generating finite element method technology has, until now, not been readily possible.

## 7   Demonstrations

### 7.1   Unknown conductivity

We start with the $L^2$ norm for our two model-data misfit functionals

$$J_{\text{misfit}}^{\text{field}} = \int\limits_{\Omega} (u_{\text{interpolated}} - u)^2 \mathrm{d}x \tag{20}$$

and

$$J_{\text{misfit}}^{\text{point}} = \int\limits_{\Omega_v} (u_{\text{obs}} - \mathcal{I}_{\text{P0DG}(\Omega_v)}(u))^2 \mathrm{d}x_v \tag{21}$$

where $u_{\text{obs}} \in \text{P0DG}(\Omega_v)$. Since integrations are equivalent to the sums of point evaluations in P0DG($\Omega_v$) (Eq. 6), the $L^2$ norm
is the same as the euclidean ($l^2$) norm. Our misfit is therefore evaluated as

$$J_{\text{misfit}}^{\text{point}} = \sum_{i=0}^{N-1} (u_{\text{obs}}^i - u(X_i))^2. \tag{22}$$

We will apply this to the simple model

$$-\nabla \cdot k \nabla u = f \tag{23}$$

for some solution field $u$ and known forcing term $f = 1$ with conductivity field $k$ under strong (Dirichlet) boundary conditions

$$u = 0 \text{ on } \Gamma \tag{24}$$

where $\Gamma$ is the domain boundary. We assert conductivity $k$ is positive by

$$k = k_0 e^q \tag{25}$$

---

[7]The current implementation of dolfin-adjoint/pyadjoint is a general AD tool for the python language using algorithms from Naumann (2011). This is described in detail in Mitusch (2018). Firedrake includes a wrapper around dolfin-adjoint/pyadjoint which allows AD of Firedrake operations such as interpolation.

[8]This requires automated formulation of adjoint PDE systems. See Farrell et al. (2013) and Sect. 1.4 of Schwedes et al. (2017) for more detail.

with $k_0 = 0.5$.

Our inverse problem is then to infer, for our known forcing field term $f$, the log-conductivity field $q$ using noisy sparse point measurements of our solution field $u$. This example has the advantage of being relatively simple whilst having a control term $q$ which is nonlinear in the model.

To avoid considering model discretisation error we generate the log-conductivity field $q_{\text{true}}$ in the space of order 2 continuous Lagrange polynomials (P2CG) in 2D on a $32 \times 32$ unit-square mesh $\Omega$ with 2048 triangular cells. We then solve the model on the same mesh to get the solution field $u_{\text{true}} \in \text{P2CG}(\Omega)$. $N$ point measurements $\{u_{\text{obs}}^i\}_0^{N-1}$ at coordinates $\{X_i\}_0^{N-1}$ are sampled from $u_{\text{true}}$ and Gaussian random noise with standard deviation $\{\sigma_i\}_0^{N-1}$ is added to each measurement.

We use a smoothing regularisation on our $q$ field which is weighted with a parameter $\alpha$. This helps to avoid over-fitting to the errors in $u_{\text{obs}}$ which are introduced by the Gaussian random noise. We now have two functionals which we minimise

$$J^{\text{point}}[u,q] = \underbrace{\int_{\Omega_v} (u_{\text{obs}} - \mathcal{I}_{\text{P0DG}(\Omega_v)}(u))^2 \mathrm{d}x_v}_{J_{\text{misfit}}^{\text{point}}} + \underbrace{\alpha^2 \int_\Omega |\nabla q|^2 \mathrm{d}x}_{J_{\text{regularisation}}} \tag{26}$$

and

$$J^{\text{field}}[u,q] = \underbrace{\int_\Omega (u_{\text{interpolated}} - u)^2 \mathrm{d}x}_{J_{\text{misfit}}^{\text{field}}} + \underbrace{\alpha^2 \int_\Omega |\nabla q|^2 \mathrm{d}x}_{J_{\text{regularisation}}}. \tag{27}$$

Each available method in SciPy's interpolation library are tested to find $u_{\text{interpolated}}$:

- $u_{\text{interpolated}}^{\text{near.}}$ using `scipy.interpolate.NearestNDInterpolator`,

- $u_{\text{interpolated}}^{\text{lin.}}$ using `scipy.interpolate.LinearNDInterpolator`,

- $u_{\text{interpolated}}^{\text{c.t.}}$ using `scipy.interpolate.CloughTocher2DInterpolator` with `fill_value = 0.0` and

- $u_{\text{interpolated}}^{\text{gau.}}$ using `scipy.interpolate.Rbf` with Gaussian radial basis function.

Note that since $u_{\text{interpolated}} \in \text{P2CG}(\Omega)$ each of 6 degrees of freedom per mesh cell has to have a value estimated given the available $u_{\text{obs}}$.

The estimated log-conductivity $q_{\text{est}}$ which minimise the functionals are found by generating code for the adjoint of our model using dolfin-adjoint/pyadjoint then using the Newton-CG minimiser from the `scipy.optimize` library. To use Newton-CG the ability to calculate Hessian-vector products for Firedrake interpolation operations was added to Firedrake's pyadjoint plugin modules.[9]

To try and balance the relative weights of the model-data misfit and regularisation terms in $J^{\text{point}}$ and $J^{\text{field}}$ we perform an L-curve analysis (Hansen and O'Leary, 1993) to find a suitable $\alpha$ following the example of Shapero et al. (2016). The L-curves

---

[9]Pyadjoint uses a forward-over-reverse scheme to calculate Hessian-vector products via an implementation of Eq. 3.8 in Naumann (2011).

were gathered for $N = 256$ randomly chosen point measurements with the resultant plots shown in Fig. A1 and Fig. A2. For low $\alpha$, $J^{\text{field}}_{\text{misfit}}$ stopped being minimised and solver divergences were seen due to the problem becoming ill formed. $\alpha = 0.02$ was therefore chosen for each method. For consistency $\alpha = 0.02$ was also used for $J^{\text{point}}$.

An extract of the Firedrake and dolfin-adjoint/pyadjoint code needed to minimise $J^{\text{point}}$ is shown in Listing 2. Once again, the Firedrake expression for $J^{\text{point}}$ in the code is the same as the mathematics in Eq. 26 given that `assemble` performs integration
over the necessary mesh. The last 3 lines are all that are required to minimise our functional with respect to $q$, with all necessary code being generated. Note that we require a reduced functional (`firedrake_adjoint.ReducedFunctional`) since the optimisation problem depends on both $q$ and $u(q)$: for a thorough explanation see Sect. 1.4 of Schwedes et al. (2017).

**Listing 2.** Firedrake code for expressing $J^{\text{point}}$ (Eq. 26) and dolfin-adjoint/pyadjoint code (inside a Firedrake wrapper) for minimizing it with respect to $q$. The omitted PDE solve code is very similar to that in Listing 1; for more see code and and data availability.

```
1: from firedrake import *
2: import firedrake_adjoint  # This is Firedrake's dolfin-adjoint/pyadjoint plugin module
3:
4: # Import our noisy samples of the true u
5: u_obs_coords = ...
6: u_obs_vals = ...
7:
8: # Solve PDE with a guess for q giving an initial u
9: ...
10:
11: omega_v = VertexOnlyMesh(omega, u_obs_coords)
12: P0DG = FunctionSpace(omega_v, 'DG', 0)
13: u_obs = Function(P0DG)
14: u_obs.dat.data[:] = u_obs_vals
15:
16: J_misfit = assemble((u_obs - interpolate(u, P0DG))**2 * dx)
17: alpha = Constant(0.02)
18: J_regularisation = assemble(alpha**2 * inner(grad(q), grad(q)) * dx)
19: J = J_misfit + J_regularisation
20:
21: q_hat = firedrake_adjoint.Control(q)
22: J_hat = firedrake_adjoint.ReducedFunctional(J, q_hat)
23: q_min = firedrake_adjoint.minimize(J_hat, method='Newton-CG')
```

### 7.1.1 Consistent point data assimilation

Our first experiment aims to check empirically that solutions of the inverse problem using point data (1) can give appreciably different results from first interpolating to the finite element mesh and (2) can have errors that are smaller than those from first interpolating to the mesh for sufficiently many observations.

It is clear that this ought to happen for $J^{\text{point}}$, the point evaluation approach, since increasing $N$ increases the number of terms in the model-data misfit sum (i.e. the integral over $\Omega_v$ gets bigger: see the equivalency of Eq. 21 and Eq. 22). There is no such mechanism for $J^{\text{field}}$, the field reconstruction approaches, since adding more data merely ought to cause $u_{\text{interpolated}}$ to approach $u_{\text{true}}$ without increasing the relative magnitude of the misfit term. The interpolation approach introduces an inconsistent treatment of the point observation errors.

Figure 2 demonstrates that the problem formulated with $J^{\text{point}}$, produces a $q_{\text{est}}$ which is closer to $q_{\text{true}}$ for all but the lowest $N$ when compared to our formulations with $J^{\text{field}}$.[10] The point evaluation approach therefore gives us *consistent* point data assimilation when compared to the particular field reconstruction approach we test.

It is possible, were an L-curve analysis repeated for each $N$, that errors in our field reconstruction approach could be reduced. The lack of convergence would not change due to there being no mechanism for growing the misfit term with number of measurements.

We could attempt to enforce consistency on the field reconstruction approach (minimising $J^{\text{field}}$) by introducing a term in the model-data misfit which increases with the number of measurements. Example calculated fields are shown in Fig. 3 and Fig. 4. These demonstrate that the choice of interpolation method changes our field reconstruction. When attempting to enforce consistency we would also need to ensure that our field reconstruction method approaches the true field as more measurement are performed. There is no obvious way to do this which is universally applicable, particularly since measurements are always subject to noise.

### 7.2 Groundwater hydrology

Our next example comes from groundwater hydrology. The key field of interest is the *hydraulic head* $\phi$, which has units of length. In the following, we will apply the Dupuit approximation, which assumes that the main variations are in the horizontal dimension.

The first main equation of groundwater hydrology is the mass conservation equation

$$S\frac{\partial \phi}{\partial t} + \nabla \cdot \mathbf{u} = q \tag{28}$$

where $S$ is the dimensionless *storativity*, $\mathbf{u}$ is the water flux, and $q$ are the sources of water. The second main equation is *Darcy's law*, which states that the water flux is proportional to the negative gradient of hydraulic head:

$$\mathbf{u} = -T\nabla\phi \tag{29}$$

---

[10]Note that we have not optimised our $\alpha$ for minimising $J^{\text{point}}$ so it is not unreasonable to assume that the prior is dominating the solution for low $N$.

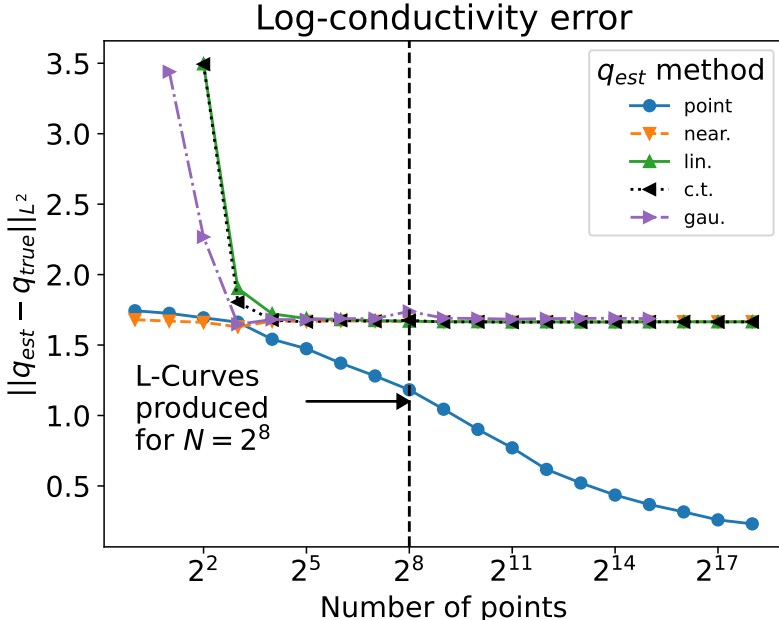

**Figure 2.** Error change as number of points $N$ is increased for minimising $J^{point}$ ($u_{interpolated}$ and $q_{est}$ method 'point' - see Eq. 26) and $J^{field}$ (the other lines - see Eq. 27) with different methods for estimating $u_{interpolated}$ where $\alpha = 0.02$ throughout (see main text for justification). The L-curves for $\alpha = 0.02$ with $N = 256$ are shown in Fig. A1 and Fig. A2. Not all methods allowed $u_{interpolated}$ to be reconstructed either due to there being too few point measurements or the interpolator requiring more system memory than was available.

where $T$ has units of length$^2$ per unit time and is known as the *transmissivity*. The transmissivity is the product of the aquifer thickness and the hydraulic conductivity, the latter of which measures the ease with which water can percolate through the medium. Clays have very low conductivity, sand and gravel much higher, and silty soil in between.

A typical inverse problem in groundwater hydrology is to determine the storativity and the transmissivity from measurements. The measurements are drawn at isolated *observation wells* where the hydraulic head can be measured directly. To create a response out of steady state, water is removed at a set of discrete *pumping wells*.

In the following, we will show a test case based on exercise 4.2.1-4.2.3 from Sun (2013). The setup for the model is a rectangular domain with the hydraulic head held at a constant value on the left-hand side of the domain and no outflow on the remaining sides. The transmissivity is a constant in three distinct zones. The hydraulic head is initially a uniform 100m and a pumping well draws 2000 m$^3$ / day towards the right-hand side of the domain. The exact value of the transmissivity and the final value of the hydraulic head are shown in Fig. 5.

For the inverse problem, the goal is to estimate the unknown values of the transmissivity in each zone, assuming that the boundaries between the zones of distinct transmissivity are known. To create the synthetic observations, we use a finite set of observation wells that take measurements at regular time intervals and perturb the exact values with normally-distributed errors

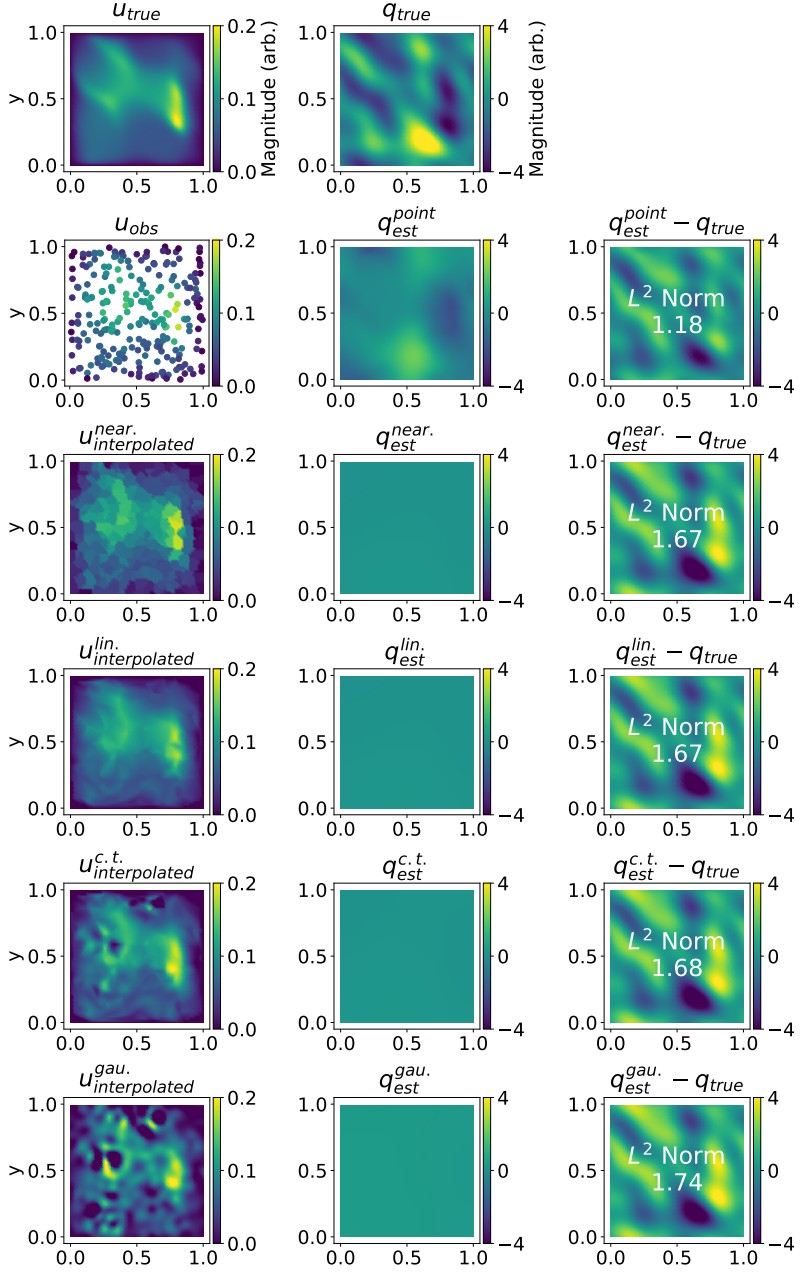

**Figure 3.** Summary plot of fields for $N = 256$. Rows correspond to method used where column 1 is the necessary $u$, column 2 is the corresponding $q$ at the optimum solution, and column 3 is the error. Row 1 shows the true $u$ and $q$. Row 2 shows the results of minimising $J^{\text{point}}$ (Eq. 26) whilst rows 3-6 show the result of minimising $J^{\text{field}}$ (Eq. 27). The regularisation parameter $\alpha = 0.02$ throughout. The field we obtain after minimising $J^{\text{point}}$, $q_{\text{est}}^{\text{point}}$, manages to reproduce some features of $q_{\text{true}}$. For minimising $J^{\text{field}}$ the solutions fail to reproduce any features of $q_{\text{true}}$ and the error is therefore higher. Each of the $u_{\text{interpolated}}$ fields are also visibly different from one another. For comparison see Fig. 4.

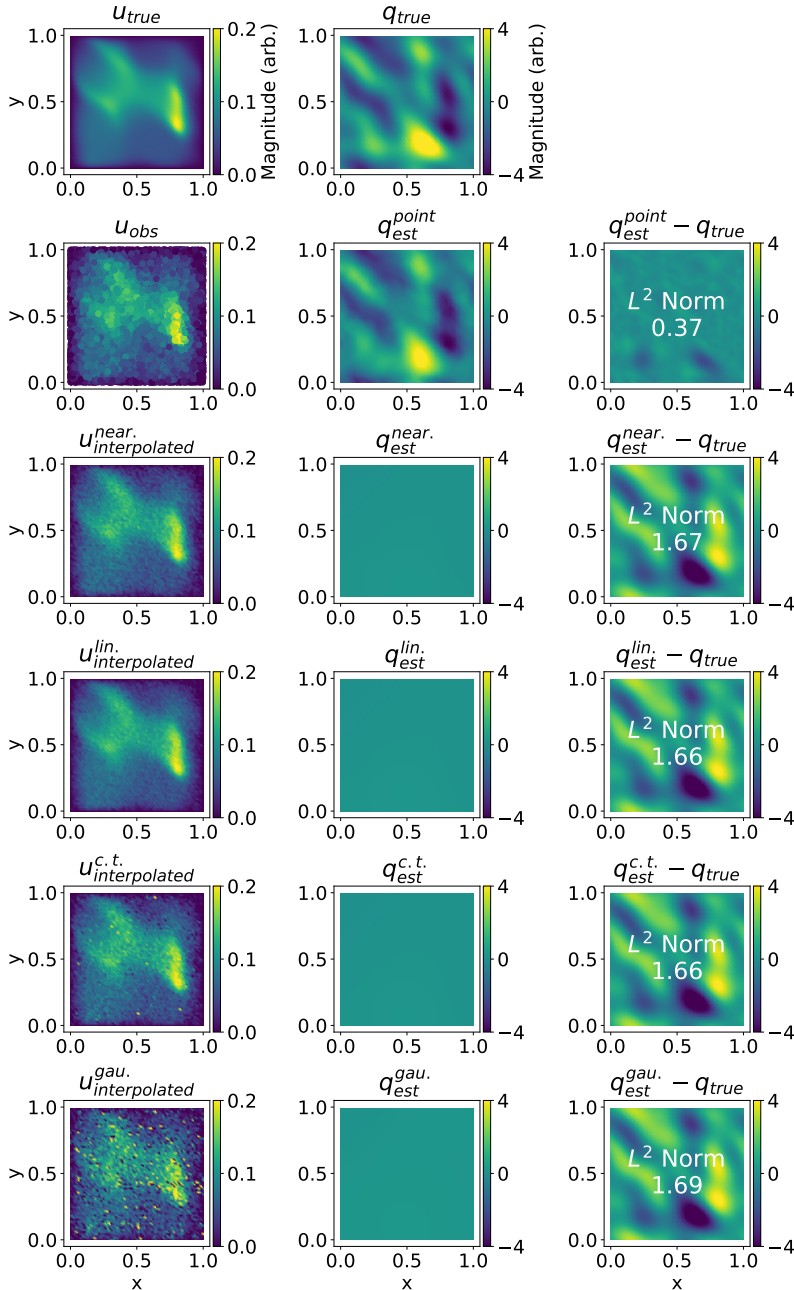

**Figure 4.** Summary plot of fields for $N = 32768$. Rows and columns correspond to those in Fig. 3. The regularisation parameter $\alpha = 0.02$ throughout. We would expect the larger number of measurements to correspondingly reduce the error: this only occurs to a significant degree when solving $J^{\text{point}}$. This cannot be entirely blamed on a lack of mechanism in $J^{\text{field}}$ for having the misfit term outgrow the regularisation term: the $u_{\text{interpolated}}$ fields do not approximate $u_{\text{true}}$ with $u_{\text{interpolated}}^{\text{gau.}}$ being particularly poor.

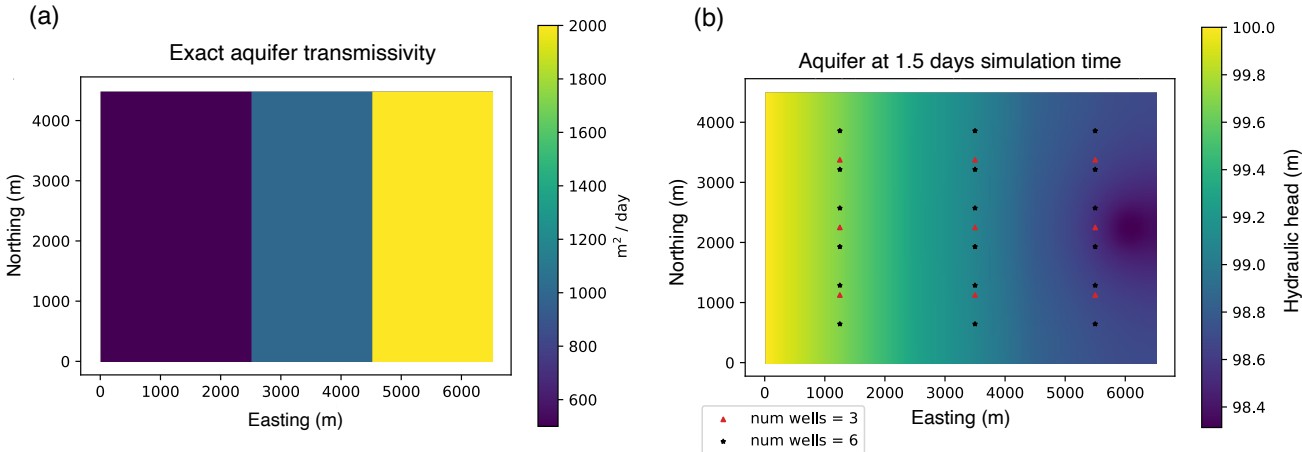

**Figure 5.** Exact transmissivity of the synthetic aquifer (a) and the final value of the hydraulic head (b).

with a standard deviation of 1 cm. In other words, the observations are

$$\phi_{kl}^{\text{obs}} = \phi(x_k, t_l) + \epsilon_{kl} \tag{30}$$

where $\{x_k\}$, $\{t_k\}$ are the observation points and times, $\epsilon_{kl}$ are the measurement errors, and $\phi$ is the true solution. We examine

two scenarios: (1) there are 6 observation wells in each zone that take measurements every 12 hours and (2) there are 2 observation wells in each zone that take measurements every 3 hours. There are three zones and thus only three parameters to infer, but we have 18 observations. Consequently, computing the maximum likelihood estimate (MLE) of the transmissivities is well-posed. We could instead think of this as computing the MAP estimate in a Bayesian inference problem using the improper uniform prior; the posterior is still proper and unimodal.

Since this is a synthetic problem, we can generate multiple statistically independent realisations of the observations $\phi^{\text{obs}}$, compute a maximum likelihood estimator from each realisation, and then explore the sampling distribution of the estimates. Here we use 30 independent realisations, for which the distribution about the mean is approximately normal. Figure 6 shows the obtained densities for the transmissivity in each zone for the two different observation scenarios. In this case, we can observe that using more observation wells but fewer measurement times resulted in a smaller variance in the inferred transmissivity

values than using the same number of total observations but with fewer observation wells and more measurement times. While this is a highly idealised problem, these kinds of experiments can inform real practice – in this case, how to balance spatial and temporal density of measurements under limited resources.

The experiment above can only be conducted when the finite element modelling API includes support for assimilating point data. In this case, the measurements are so sparse that they cannot be meaningfully interpolated to a densely-defined field.

Nonetheless, we can still compute a maximum likelihood estimate for the unknown parameters.

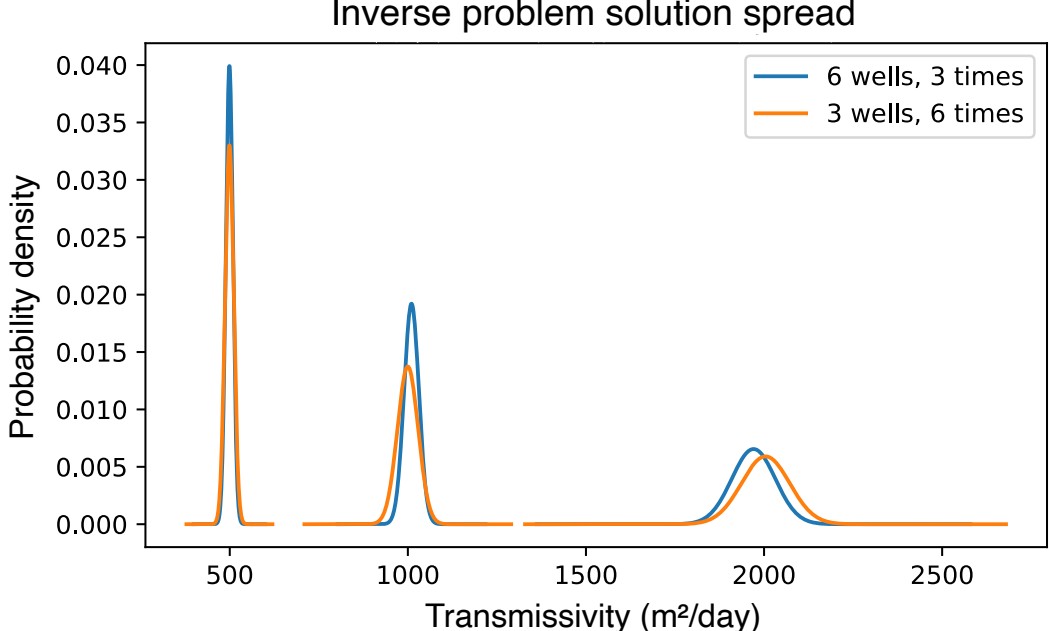

**Figure 6.** Probability densities for the inferred transmissivities in each zone. The blue curves show the results obtained with a larger number of observation wells but fewer measurement times, while the orange curves show fewer observation wells but more measurement times.

### 7.3 Ice shelves

#### 7.3.1 Physics model

Our final example comes from glaciology; the main field of interest is the ice velocity. On length and time scales greater than 100m and several days, glaciers flow like a viscous fluid. The most principled equation set determining the velocity of a viscous

fluid are the full Navier-Stokes equations, but the equation set we will work with uses several simplifications. First, ice flow occurs at very low Reynolds number, and the ratio of the thickness to the length of the spatial domain is usually on the order of 1/20 or less. Second, we will focus on *ice shelves* – areas where a glacier floats on the open ocean. Most of the drainage basins of the Antarctic Ice Sheet terminate in floating ice shelves. As far as the dynamics are concerned, ice shelves experience almost no friction at their beds. As a consequence, the horizontal velocity is nearly constant with depth, so we can depth-average the

equations. The resulting PDE is called the *shallow shelf equations*, which we describe below. For complete derivations of all the common models used in glacier dynamics, see Greve and Blatter (2009).

     The main unknown variable in the shallow shelf equations is the depth-averaged ice velocity $\mathbf{u}$, which is a 2D vector field. The other key unknown is the ice thickness $h$. Since an ice shelf is floating on the ocean, by matching the pressures at the base of the ice we find that the surface elevation of an ice shelf is $s = (1 - \rho_I/\rho_W)h$ where $\rho_I$, $\rho_W$ are respectively the density of

ice and ocean water.

A key intermediate variable is the *membrane stress tensor*, which we will write as $\mathbf{M}$. The membrane stress tensor has units of stress and has rank 2, i.e. it is a $2 \times 2$ matrix field. Physically, the membrane stress plays the same role for this simplified 2D problem as the full stress tensor does for the full Stokes equations. The shallow shelf equations are a conservation law for membrane stress:

$$\nabla \cdot (h\mathbf{M}) - \frac{1}{2}\rho_I(1 - \rho_I/\rho_W)g\nabla h^2 = 0 \tag{31}$$

where $g$ is the acceleration due to gravity.

To obtain a closed system of equations, we need to supply a *constitutive relation* – an equation relating the membrane stress tensor to the depth-averaged velocity. First, the *strain rate tensor* is defined to be the rank-2 tensor

$$\dot{\varepsilon} = \frac{1}{2}\left(\nabla\mathbf{u} + \nabla\mathbf{u}^\top\right), \tag{32}$$

i.e. the symmetrized gradient of the depth-averaged velocity. The membrane stress tensor is then proportional to the strain rate tensor:

$$\mathbf{M} = 2\mu\left(\dot{\varepsilon} + \operatorname{tr}(\dot{\varepsilon})\mathbf{I}\right) \tag{33}$$

where $\mathbf{I}$ is the $2 \times 2$ identity tensor and $\mu$ is the viscosity coefficient, which has units of stress $\times$ time.

One of the more challenging parts about glacier dynamics is that the viscosity also depends on the strain rate. This makes the shallow shelf equations nonlinear in the velocity. The most common assumption is that the viscosity is a power-law function of the strain rate tensor:

$$\mu = \frac{A^{-\frac{1}{n}}}{2}\sqrt{\frac{\dot{\varepsilon} : \dot{\varepsilon} + \operatorname{tr}(\dot{\varepsilon})^2}{2}}^{\,\frac{1}{n}-1} \tag{34}$$

where $n$ is an exponent and $A$ is a prefactor called the *fluidity*. Laboratory experiments and field observations show that $n \approx 3$; this is referred to as *Glen's flow law* (Greve and Blatter, 2009). The fact that $n > 1$ makes ice a *shear-thinning* fluid, i.e. the resistance to flow decreases at higher strain rate. The fluidity coefficient $A$ has units of stress$^n$ $\times$ time. (This unit choice and the exponent of $-1/n$ on $A$ in equation (34) reflects the fact that historically glaciologists have, by convention, written the constitutive relation as an equation defining the strain rate as a function of the stress. For solving the momentum balance equations, we have to invert this relation.) Several factors determine the fluidity, the most important of which is temperature – warmer ice is easier to deform.

Putting together equations (31), (32), (33), (34), we get a nonlinear second-order elliptic PDE for $\mathbf{u}$. The last thing we need to complete our description of the problem is a set of boundary conditions. We fix the ice velocity along the inflow boundary, i.e. a Dirichlet condition. Along the outflow boundary, we fix the normal component of the membrane stress:

$$h\mathbf{M} \cdot \boldsymbol{\nu} = \frac{1}{2}\rho_I(1 - \rho_I/\rho_W)gh^2\boldsymbol{\nu} \tag{35}$$

where $\boldsymbol{\nu}$ is the unit outward-pointing normal vector to the boundary of the domain. This is a Neumann-type boundary condition.

## 7.3.2 Inverse problem

The key unknowns in the shallow shelf equations are the thickness, velocity, and fluidity. The ice velocity and surface elevation are observable at large scales through satellite remote sensing. Assuming that a floating ice shelf is in hydrostatic equilibrium, the thickness can then be calculated from the surface elevation given the density of ice and seawater and some estimate for the air content in the snow and firn on top of the ice. The fluidity, on the other hand, is not directly measurable at large scales. The goal of our inverse problem is to estimate the fluidity from measurements of ice velocity.

As a test case, we will look at the Larsen C Ice Shelf in the Antarctic Peninsula. We will use the BedMachine map of Antarctic ice thickness (Morlighem et al., 2020) and the MEaSUREs InSAR phase-based velocity map (Mouginot et al., 2019).

To ensure positivity of the fluidity field that we estimate, we will, as in Sect. 7.1, write

$$A = A_0 e^{\theta} \tag{36}$$

and infer the dimensionless log-fluidity field $\theta$ instead.

The first paper to consider inverse problems or data assimilation in glaciology was MacAyeal (1992) which referred to the formulation of a functional to be minimised as the *control method*. Since then, most of the work in the glaciology literature on data assimilation has assumed that the observational data can be interpolated to a continuously-defined field and used as a misfit functional the 2-norm difference between the interpolated velocity and the computed velocity (Joughin et al., 2004; Vieli et al., 2006; Shapero et al., 2016). Assuming that the target velocity field to match is defined continuously throughout the entire domain, however, obscures the fact that there are only a finite number of data points. A handful of publications have taken the number of observations into account explicitly in order to apply further statistical tests on goodness-of-fit, for example MacAyeal et al. (1995). We argue that making it possible to easily assimilate sparse data will improve the statistical interpretability of the results. Moreover, one of the main uses for data assimilation is to provide an estimate of the initial state of the ice sheet for use in projections of ice flow and extent into the future. Improving the statistical interpretability of the results of glaciological data assimilation will help to quantify the spread in model projections due to uncertainty in the estimated initial state.

Virtually all existing work on glaciological inverse problems assumes that the measurement error variance $\sigma$ is known. These errors are estimated by experts in remote sensing, but different data products use different methodologies for error estimation. For example, the documentation for the Inter-mission Time Series of Land Ice Velocities (ITS_LIVE) states explicitly that their reported errors are "unrealistically low" (Gardner et al., 2019). If the only goal is to compute a single best estimate of an unknown field, then the provided error estimates can be used as a qualitative weighting which encodes the fact that the velocity is better estimated in some regions than others. Under some assumptions, it is possible to get a better sense of what the true errors are. The assumptions are

1. The physics model relating the unknown parameters and observable fields is correct.

2. The errors $\sigma_k$ that the observationalists provide are correct in a relative sense but not an absolute sense. To be precise, we assume that for any two measurements $\sigma_j$ and $\sigma_k$, the ratio $\sigma_k/\sigma_j$ is known, but the exact magnitude of $\sigma_j$ or $\sigma_k$ is not.

Our goal is then to estimate not just the unknown parameters $\theta$, but also a uniform scaling factor $c$ such that the true measurement errors $\hat{\sigma}$ can be written as

$$\hat{\sigma} = c\sigma. \tag{37}$$

We note that these assumptions might not be justified. For example, our ice flow model might be inadequate. Alternatively, the true measurement errors might not even be correct in a relative sense, in which case the simple relation (37) does not hold. Nonetheless, if we do make these assumptions, we can estimate the scaling parameter $c$ through a *cross-validation* experiment.

The idea of cross-validation is to use only a subset of the observational data $u^o$ to estimate the unknown field. The data that are held out are then used to determine the goodness of fit. Rather than use, for example, the Morozov discrepancy principle (Habermann et al., 2012) or the L-curve method to select the regularisation parameter, we can choose it as whichever value gives the smallest misfit on the held-out data. Crucially, the total misfit on the held-out data can then be used to estimate the error scaling factor $c$. This experiment would be meaningless using the field reconstruction approach because, at points where we have held out data, we would be matching the computed velocity field to interpolated values, when the whole point of the exercise is to not match the computed velocity field to anything where we have no data. A common approach is to leave out only a single data point; for linear problems, there are analytical results that allow for much easier estimation of the best value of the regularisation parameter using leave-one-out cross-validation (Picard and Cook, 1984). Here we will instead randomly leave out some percentage of the observational data.

Formally, let $\{u^o(x_k)\}$ be a set of $N$ observations of the velocity field at points $\{x_k\}$. Let $f$ be some parameter between 0 and 1 and select uniformly at random a subset $I$ of size $f \times N$ of indices between 1 and N. The model-data misfit functional for our problem is

$$E(u) = \sum_{k \in I} \frac{|u(x_k) - u^o(x_k)|^2}{2\sigma_k^2} \tag{38}$$

where $\sigma_k$ is the formal estimate of the error of the $k$th measurement as reported from the remote sensing data. Note that we sum over only the data points in $I$ and not all of the observational data. The regularisation functional is, again similarly to Sect. 7.1,

$$R(\theta) = \frac{\alpha^2}{2} \int_\Omega |\nabla \theta|^2 \mathrm{d}x. \tag{39}$$

We can then minimise the functional

$$E(u) + R(\theta) \tag{40}$$

subject to the constraint that $u$ is a solution of the momentum balance equations 31, 32, 33, 34 for the given $\theta$.

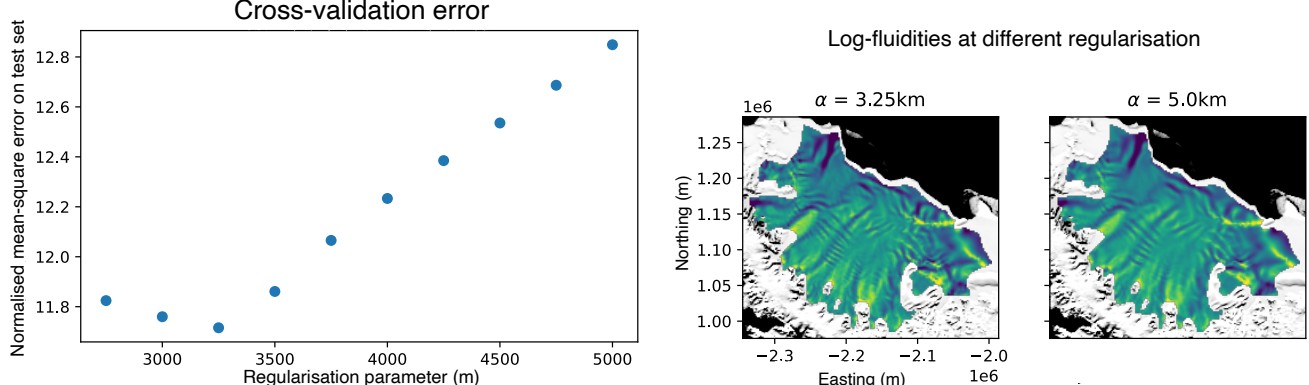

**Figure 7.** Normalized cross-validation errors ($E'(\alpha)$) and inferred log-fluidity fields with regularisation parameter set to 3.25 km and 5km. The log-fluidity background image is the MODIS Mosaic of Antarctica (Scambos et al., 2007; Haran et al., 2021), courtesy of the NASA National Snow and Ice Data Center (NSIDC) Distributed Active Archive Center (DAAC).

We now let $\theta_\alpha$, $u_\alpha$ be the log-fluidity and velocity obtained by minimising Eq. 40 using a regularisation parameter $\alpha$. We will then compute the misfit against the held-out data:

$$E'(\alpha) = \sum_{k \notin I} \frac{|u_\alpha(x_k) - u^o(x_k)|^2}{2\sigma_k^2}. \tag{41}$$

The key difference here is that we sum over the indices *not* in the training set ($k \notin I$) instead of those in the training set ($k \in I$). The right choice of $\alpha$ is the minimiser of $E'$.

This experiment was performed using Icepack (Shapero et al., 2021), which is built on top of Firedrake. As our target site, we used the Larsen C Ice Shelf in the Antarctic Peninsula. There are a total of 235,510 grid points within the domain; we used 5% or 11,775 of these as our training points. [11] To pick the right value of $\alpha$, we used a regular sampling from 2.5km to 5km. The results of the cross-validation experiment are shown in Fig. 7. We found a well-defined minimum at $\alpha = 3.25$ km. Fig. 7 shows the log-fluidity fields obtained at 3.25 km, which is the appropriate regularisation level, and 5 km, which is over-regularised. With $\alpha = 5$ km, several features are obscured or blurred out.

Having found the optimal value of the regularisation parameter, we can now estimate the scaling factor $c$ in equation (37). The normalised sum of squared cross-validation errors of the regularisation parameter was roughly 11.8, which suggests that the formal errors under-estimate the true errors by a factor of about 3.4 by taking square roots.

To summarize, this experiment demonstrates the new point data assimilation features of Firedrake on a problem using real observational data. The cross-validation approach that we use here is one possible alternative to other methods for picking the regularisation parameter, such as the L-curve or Morozov discrepancy principle. Cross-validation is only possible with point

---

[11]Note that the gridded velocity data are the result of several steps of post-processing and interpolation from an unstructured point cloud of raw displacements. The raw displacement are obtained via repeat-image feature tracking but are not usually made available; an ideal version of this experiment would use this point cloud instead of the gridded velocity data.

data assimilation as opposed to interpolating the data and fitting to the interpolated field. An advantage of cross-validation is that, under certain assumptions, it can provide an independent estimate of the standard deviation of the measurement errors. Pinning down the degree to which the formal errors under- or over-estimate the true errors is difficult by any other means. The Morozov discrepancy principle, for example, assumes that the measurement errors used in the objective functional are the true measurement errors (Habermann et al., 2012), and breaks down when this assumption is violated. Finally, although fitting to point data is an improvement to the standard approaches used in glaciological data assimilation, many further improvements are possible on the problem formulation shown here. For example, a formal Bayesian treatment would be able to address uncertainty quantification in a way that the approach used here cannot.

## 8 Future Point Data Extensions

The representation of point data as a function in a finite element space, and its implementation in Firedrake, provide the basis on which a range of further abstractions and functionality might be built.

A clear limitation of the work presented is that the points are static. A logical next step would be toœ implement moving points. The underlying concepts of interpolating from a parent mesh onto a vertex-only mesh remain unchanged: the vertex-only mesh would now move over time. An unresolved question is the optimal differentiable abstraction for particle movement. In particular, should the differentiation go inside or outside the ODE solver used to move the particles?

Moving points could be used for assimilating data from Lagrangian points, such as data from weather buoys which follow ocean currents. For static experiments, we could include uncertainty in the location of measurements through a differentiable point relocation operator. Moving points could allow particle in cell methods to be introduce to Firedrake as has been done for FEniCS (Maljaars et al., 2021).

The functionality presented here has enabled cross-mesh parallel compatible interpolation between meshes in Firedrake.[12] This provides users with numerous data analysis possibilities including the ability to compute arbitrary flux and line integrals of solution fields. It also has the potential to be employed to assimilate data produced by models using other (not finite element) discretisations.

More generally, the fact that any dual functional can be approximated by a quadrature rule could be exploited to create a general model coupling capability. Where both models are written in Firedake, this could even be extended to support tight coupling in which implicit linear operators span the coupled components.

## 9 Conclusions

This work makes two key points. First, a finite element representation of point data enables the automation of point evaluations. This feature then makes it possible to solve PDE-constrained optimisation problems where the objective functional contains point evaluations. Second, assimilating point data directly instead of using field reconstruction has several benefits: it (1) avoids

---

[12]Prior work on supermeshing (Maddison and Farrell, 2012; Farrell et al., 2009) provides a mathematical framework.

the need for ambiguously defined inter-measurement interpolation, (2) allows assimilation of very sparse measurements, and (3) enables more in-depth statistical analysis. Moreover, using point data directly is beneficial whether one views the procedure

as solving a deterministic inverse problem or as a component of Bayesian inference.

The second result is both a demonstration of new functionality and a general call for all scientific communities who face these kinds of inverse problems to carefully consider if point evaluation misfit functionals would be appropriate for their use case. This is particularly salient for users of finite element methods where point evaluation of fields is always well defined over the whole domain. For data assimilation problems where this is not possible, ensuring that the model-data misfit grows with

the number of measurements could also be considered.

**Appendix A:  L curves**

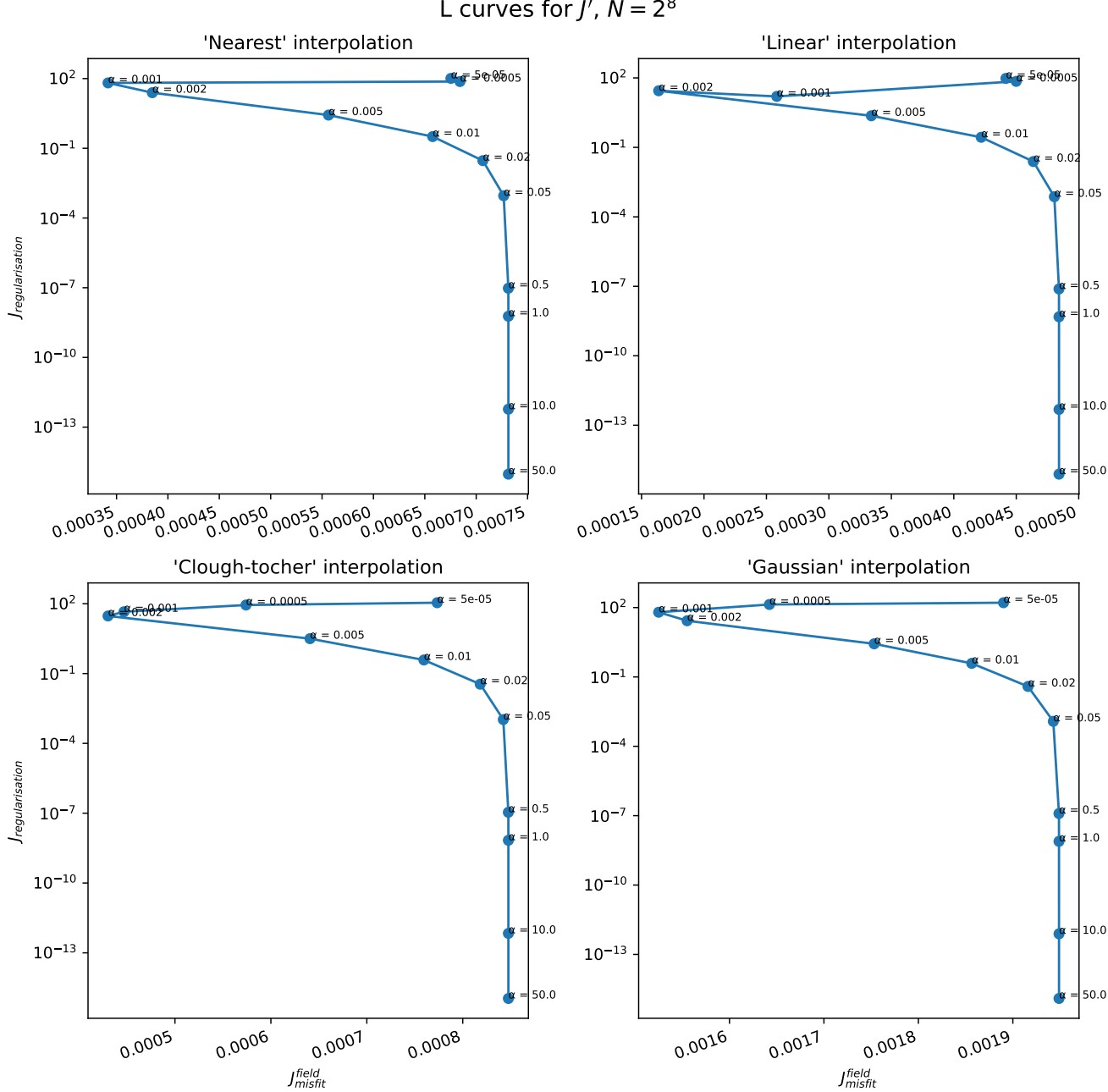

**Figure A1.** L-curves from minimising $J^{\text{field}}$ (Eq. 27) with different methods for estimating $u_{\text{interpolated}}$. For low $\alpha$, $J^{\text{field}}_{\text{misfit}}$ stopped being minimised and solver divergences were seen for $u^{\text{gaussian}}_{\text{interpolated}}$. The problem was likely becoming overly ill formed and the characteristic 'L' shape (with sharply rising $J_{\text{regularisation}}$ for low $\alpha$ and a tail-off for large $\alpha$) is therefore not seen. To keep the problem well formed without the regularisation parameter being too big $\alpha = 0.02$ is chosen for each method.

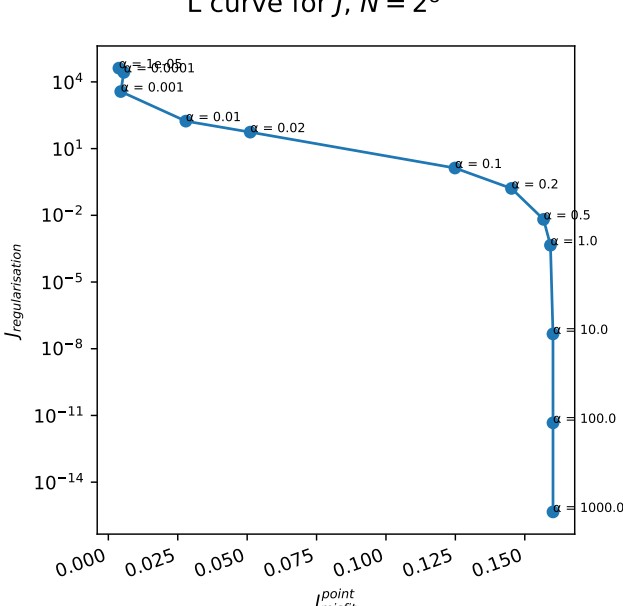

**Figure A2.** L-curve for minimising $J^{\text{point}}$ (Eq. 26). The characteristic 'L' shape is seen and $\alpha = 0.02$ is seen to be close to the turning point and is therefore chosen for consistency with the other L-curves (Fig. A1).

*Code and data availability.* The demonstrations use publicly available code in a GitHub repository which is archived with Zenodo (Nixon-Hill and Shapero, 2023). All figures can be generated using that repository. The version of Firedrake used for the unknown conductivity demonstration is archived on Zenodo (zenodo/Firedrake-20230316.0). The version of Firedrake used for the groundwater hydrology and ice shelf demonstration is similarly archived (zenodo/Firedrake-20230405.1). The ice shelf demonstration uses Icepack (Shapero et al., 2021), specifically this Zenodo archived version (Shapero et al., 2023).

The BedMachine thickness map (Morlighem et al., 2020; Morlighem, 2022) and the MEaSUREs InSAR phase-based velocity map (Mouginot et al., 2019) used in the ice shelf demo are publicly hosted at the US National Snow and Ice Data Center.

*Author contributions.* RWNH and DAH formulated the point data point evaluation ideas with CJC validating the mathematics. RWNH made software modifications to Firedrake (Rathgeber et al., 2016), FInAT (Homolya et al., 2017), FIAT (Kirby, 2004), UFL (Alnæs et al., 2014) and TSFC (Homolya et al., 2018) necessary for this work under the supervision of DAH. DS and RWNH were responsible for the unknown conductivity experiment methodology and validation; RWNH performed the experiment and was responsible for visualisation. DS was responsible for methodology of the groundwater hydrology and ice shelf experiments with RWNH performing validation; DS and RWNH were responsible for visualisations. DS is the author of the Icepack (Shapero et al., 2021) software package. RWNH and DS wrote the initial draft presentation. All authors contributed to review and editing.

*Competing interests.* David A. Ham, a coauthor of this work, is the chief-executive editor of the Geoscientific Model Development journal.

*Acknowledgements.* This work was supported by the Natural Environment Research Council [NE/S007415/1]. DRS is funded by the US National Science Foundation (grant #1835321) and National Aeronautics and Space Administration (grant #80NSSC20K0954).

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
