# Peer review of "Consistent Point Data Assimilation in Firedrake and Icepack"

_EGUsphere, 2023_

## Referee Comment (RC1)

**Review of 'Consistent Point Data Assimilation in Firedrake and Icepack' by Nixon-Hill et al.**

Doug Brinkerhoff

July 28, 2023

**1 Summary and Main Points**

This paper introduces methods in Firedrake for computing cost functions involving pointwise rather than integral likelihoods. The authors demonstrate this capacity on examples from heat conduction, groundwater hydrology, and glaciology.

I generally find the problem that this paper addresses to be an important one, and the PDE-constrained optimization community would be well served by adopting some of the reasoning contained here. I also think that this method is unfortunately muddled by artificial rigor: my chief complaint about this work is that involves more technical detail than is required to adequately address the issue at hand, and in some cases the presentation of that detail contains mistakes, although I do not think that those mistakes necessarily translate into the implementation of the work. I think that this paper can definitely be made suitable for publication in GMD, but requires some significant reframing with respect to how the methods and results are presented.

**2 Line-by-line comments**

**P1** I don't think that 'Partial Differential Equation' should be capitalized.

**P2, Para 2** 'misfits which' → 'misfits that'

**P2, Para 2** I'm not sure what 'finding a first or second derivative of the point evaluation operation' means. I suspect it means that the output of the interpolation operation needs to be amenable to reverse-mode automatic differentiation, but this should be clarified.

**Sec. 2** This section is probably a bit too textbook for GMD. I do think it's relevant to remind the reader that finite elements are functions defined everywhere on the domain, and even that they take the form of a weighted sum of basis vectors, but the basic example given makes that point less clear. In the last line of this section, I think it would be better to say 'within the boundaries of the mesh' rather than 'on the mesh'.

**Sec. 3** I think that this section significantly overcomplicates the situation. The evaluation of a finite element function at specific locations is just

$$\mathbf{u}_{\mathrm{pts}} = \mathbf{\Phi}_{\mathrm{pts}}\mathbf{w} \tag{1}$$

where $\mathbf{u}_{\mathrm{pts}}$ are the interpolated model predictions, $\mathbf{w}$ are the DoF values (presumably determined from solving a PDE), and $\mathbf{\Phi}_{\mathrm{pts}}$ is a sparse matrix with each desired evaluation location as a row, and each FEM basis function a column. This requires no definitions of vertex-only meshes and and does not require pointwise DG spaces. Indeed, such a formulation doesn't actually work: the spatial integral of a finite valued basis function that is only defined at a single point in $\mathbf{R}^d$ (as in Eq. 7) would be zero: the basis function that accomplishes the intended goal is not pointwise constant but rather the Dirac delta function. The simpler discrete linear operator viewpoint also has a particularly simple adjoint, which is, well, the adjoint (i.e. the transpose). I recognize that the vertex-only mesh may be a convenient way to represent the situation in UFL, but I think that this should be justified from a data structure perspective rather than a mathematical one.

**Sec. 4** More or less the same comments as above. I don't understand the need for the complexity here: the Ciarlet definition of a finite element method is useful for defining elements, but Firedrake already abstracts away the evaluation of functions defined over the resulting spaces (see for example the .at(x) functionality).

**Eq. 17** It would be more common to write the complete discrete variational problem by prepending the phrase: 'Find $u \in P2CG(\Omega)$ such that'

**On P9, regarding pointwise evaluation of DG functions** It seems extremely unlikely that real-valued coordinates should fall on mesh boundaries. If for some reason they do, perhaps it would be better to either throw an error or take an average.

**P9, last paragraph** I again want to emphasize that the interpolation operation reduces to a sparse matrix-vector product.

**Eq. 19** what does $i$ mean in this equation? Should these points be elements of a set?

**Just after Eq. 20** The regularization isn't a 'guess' (a word which might reasonably be used in this context to describe an initial parameter value for an iterative optimization procedure), but rather encodes assumptions about the function space in which the parameter lives.

**Note on nomenclature** This is not necessary. Simply define terms as they're used here.

**Eq. 21** To be clear, I am strongly supportive of the viewpoint the authors take here, and it is surely better to map to point observations rather than use interpolation to produce an intermediate value. However: I think that this formulation is a straw-man because it ignores the fact that different observations can be weighted differently (or likelihoods can be heteroscedastic, if you prefer). For example, if one used a Gaussian Process to interpolate point data to the computational mesh, and used the associated posterior distribution as the likelihood function (the negative log-likelihood would be the misfit functional in this formulation), then one could also use the posterior covariance as a weight, which would yield large weights (small variances) for locations on the mesh close to the point observations and small weights (large variances) elsewhere. Of course this still involves the introduction of a model controlling observational smoothness – which is probably undesirable – but it is not nearly so awful as suggested here.

**Last line of Sec. 6** Have a look at this paper: https://arxiv.org/abs/2303.06871. Several other works in recent years have coupled FEM solvers (including Firedrake) to general AD tools.

**Between Eqs. 24 and 25** Again, I don't think this is correct for DG0 on points. The basis functions would need to be Dirac delta functions rather than finite constants. However, Eq. 25 is still correct regardless.

**P13** 'L-curve' is usually capitalized.

**Sec. 7.1.1** The lack of posterior consistency in this case is the result of not actually treating the interpolation in a properly Baysesian way by ignoring the resulting posterior uncertainties! It is certainly true that treating that interpolation operator in a way that ensures that consistency is non-trivial and so the current method is still of great utility because it avoids that necessity (more or less). However, to say that there isn't a way to ensure convergence in the infinite data limit for the interpolated case isn't true.

**Eq. 32** I think that there's a units mismatch here: the time derivative is L/T, while a spatial derivative of a velocity is a 1/T. Should those velocities be fluxes?

**P18, paragraph 3** I don't understand where the probability density functions come from: this usually requires the use of Bayes' theorem and MCMC or variational inference or something to come up with such distributions. Can you describe in more detail how these were produced with the stated 'ensembles'?

**P18, last paragraph** Please see comment on Sec. 7.1.1.

**Sec. 7.3.1** I don't think it's necessary to justify use of the SSA here.

**Sec. 7.3.2** The thickness is not observable from satellite remote sensing, the surface elevation is.

**P21, para 3** One challenge here is that velocities themselves are a gridded product and may not represent the 'real' data points (whatever that means for cross-correlation methods). It would be worthwhile to mention this.

**Fig. 8** What does the 'm' in the axis label refer to?

**P23 L1** What would happen if your test set wasn't randomly sampled? Velocity observations tend to be very close together, often much closer than the characteristic smoothing scale induced by the ice sheet model (typically understood as 6 to 10 ice thicknesses), and this spatial autocorrelation would lead to an underestimate of the regularization parameter because any roughness in $\theta$ below that smoothing scale would be averaged by the physics themselves.

**P23 Para 2** I don't understand the reasoning here, nor the assumptions about uniform scaling factors. All you are measuring is the RMSE, which could be attributed either to observation errors *or* to model inadequacy.

**P23 Para 3 and 4** I don't really understand what this part is trying to say. There are many other mechanisms for doing principled inverse problems than the ones described here (dare I say that adopting Bayesian formalism and the accompanying methods are quite a bit more principled than what is described in this work). In any case, this is better suited for an independent 'Discussion' section, as it is not strictly relevant to the results of the glaciological experiment.

**Sec. 8** I don't see how different observations in time is linked to particle in cell methods, except for the fact that they both involve a 0-D object and some notion of time. In general, this section doesn't seem fully considered and the choices of what to describe are kind of arbitrary and not immediately relevant to addressing the shortcomings of the work presented here.

---

## Author Comment (AC2)

**Response to "Review of 'Consistent Point Data Assimilation in Firedrake and Icepack' by Nixon-Hill et al." by Umberto Villa**

Reuben W. Nixon Hill, Daniel Shapero, Colin J. Cotter, David A. Ham

February 16, 2024

**1 Summary**

We would like to thank the reviewer for his careful consideration of our manuscript. A number of the points he raises are very well-made and we have made significant changes to our revised manuscript as a result.

The reviewer's reason for recommending reject is that he feels that the software aspects of the paper are the key interesting component and that these should be the focus of a performance-oriented submission to a software journal such as SoftwareX or TOMS. GMD is, of course, itself a software journal, with a focus on software applicable to geoscientific applications. We therefore don't feel that it's necessary to reorient the paper onto what TOMS or SoftwareX would contain. Conversely, the other reviewer felt that the software aspects were over-detailed but that the message about data assimilation in cryosphere modules was essentially valuable, albeit with reservations about specific matters.

What we can take on board is that the manuscript is insufficiently explicit in what the reader should take away. In short, the core idea of this manuscript is that point evaluation can be fully integrated as a first class, differentiable operation in a symbolic finite element framework such as Firedrake, and that doing so makes it straightforward to assimilate point data by interpolation, as opposed to the (mathematically problematic) extrapolation methods that have frequently been applied. We will make this explanation much more explicit in the abstract and introduction.

**2 Response to Specific Comments**

1. We will cite hIPPYlib and explain the difference between that work and the approach presented here. We are not the first to assimilate point data and we do not claim to be – oceanographers have assimilated data from drifting buoys into circulation models for years. The key distinction between this and previous work such as hIPPYlib is that the point data is included as fully first class objects in the symbolic layer of the finite element system. This enables data assimilation, which hIPPYlib also supports, but also enables point data to be used almost anywhere in a calculation where any other finite element function could be used. For example, this enables the interpolation to point data to be put in the integrand, as shown in equation 13. The beauty of introducing a fully composable first class operator is that users will certainly find uses for this functionality that we have not thought of.

2. The goal of this experiment is to compare the approach the we propose with what is commonly done in the literature. Many publications have used naive interpolation methods to map discrete observations to a finite element mesh without acknowledging the impact that this arbitrary algorithmic choice can make. We agree that using Gaussian process regression would be superior to naive interpolation, but relatively few publications (at least in the glaciology literature) use it.

3. The reviewer correctly points out several instances where we overstated the statistical rigour of our experiment. We have walked back some of these claims. The purpose of the demonstrations here is to show that the new point data assimilation capabilities of Firedrake work as advertised and that this feature is fully integrated into the rest of the symbolic capabilities of the library. We claim that this new feature is useful whether one is interested in solving problems from the viewpoint of both Bayesian statistics or deterministic inverse problems.

1. We agree, we overstated our case here and have altered the text accordingly.

2. This was an oversight on our part and we thank the reviewer for pointing this out. Using the square norm of the gradient is an inverse crime and we have updated the text to reflect this. Nonetheless, the essential point of this exercise is to demonstrate the new point data assimilation capabilities added to Firedrake and that the resulting estimates converge in the limit of a large number of observations. Using the square norm of the second derivative is better practice but does not substantially alter the essential point, so we have kept the simulations as-is.

3. We have removed discussion of a Bayesian inverse problem.

4. Please see our answer to reviewer Brinkhoff about Sec. 7.1.1.

5. We have expanded on the explanation in the text. We are not using a Kalman filter or the Laplace approximation. Briefly, there are only three parameters to infer and 18 observations, so computing a MAP estimator with the improper uniform prior on the transmissivities is the same as computing the maximum likelihood estimator. We generated 30 independent realisations of the observation set, computed a MAP/MLE from each one, and then fit a normal distribution to the resulting 30 estimates for the transmissivities.

6. The reviewer suggests that scaling experiments are necessary. We suggest that this is not actually valuable in the context of the functionality presented here. The reason is that point evaluation at static points typically accounts for a vanishing proportion of runtime. The circumstances where the performance of point evaluation are likely to become a first order concern are where there are a very large number of particles and these move (hence necessitating frequent updates of the containing cell and local coordinates). This functionality is not presented here and is listed in the future work section. A future paper focussed on, for example, statistics of moving particles would be the appropriate juncture to study the performance of the system.

7. 1. We have standardised the notation to use $J^{\text{point}}$ and $J^{\text{field}}$

   2. This is a well-made point. We have pared back this section significantly.

   3. We have tightened up the formality of the mathematical exposition throughout. We would like to point that the semicolons are not typos but are an indication that the operator in question is linear in the arguments after the semicolon. This notation was introduced at the end of chapter 3 and is an extension of the notation for forms used in, for example, Alnæs et al. (2014). The use of tilde to indicate quantities in local coordinates has been made consistent.

**References**

Alnæs, M. S., Logg, A., Ølgaard, K. B., Rognes, M. E., and Wells, G. N.: Unified form language: A domain-specific language for weak formulations of partial differential equations, ACM Transactions on Mathematical Software, 40, 9:1–9:37, https://doi.org/10.1145/2566630, 2014.

---

## Author Response (AR1)

**Response to the reviews of "Consistent Point Data Assimilation in Firedrake and Icepack' by Nixon-Hill et al."**

Reuben W. Nixon Hill, Daniel Shapero, Colin J. Cotter, David A. Ham

March 25, 2024

**1   Summary**

We would like to thank both reviewers for their careful consideration of our manuscript. A number of the points raised are very well-made and we have made significant changes to our revised manuscript as a result.

Before we turn to the specific comments of each of the reviewers, we should address the concerns of the two reviewers about the overall message of the paper. Doug Brinkerhoff writes:

> I generally find the problem that this paper addresses to be an important one, and the PDE-constrained optimization community would be well served by adopting some of the reasoning contained here. I also think that this method is unfortunately muddled by artificial rigor: my chief complaint about this work is that involves more technical detail than is required to adequately address the issue at hand, and in some cases the presentation of that detail contains mistakes, although I do not think that those mistakes necessarily translate into the implementation of the work.

conversely, Umberto Villa writes:

> given the fact that in general the use of point-wise observations is well understood by the inverse problem and data assimilation communities, I am wondering if the manuscript should be more focused on the true (and remarkable) contribution that is the implementation of point-wise observation within UFL/Firedrake. As such, a manuscript focused on the implementation details and algorithms as well as the assessment of its numerical performances may be better suited to a software journal such as SoftwareX or TOMs.

One reviewer finds the data assimilation approaches presented in the paper valuable but feels the software aspects are unnecessary detail, while the other reviewer finds the software aspects "remarkable" but feels the data assimilation aspects are well-known.

What we can take on board is that the manuscript is insufficiently explicit in what the reader should take away. In short, the core idea of this manuscript is that point evaluation can be fully integrated as a first class, differentiable operation in a symbolic finite element framework such as Firedrake, and that doing so makes it straightforward to assimilate point data by interpolation, as opposed to the (mathematically problematic) extrapolation methods that have frequently been applied. We have made this explanation much more explicit in the abstract and introduction.

Umberto Villa's reason for recommending reject is that he feels that the software aspects of the paper are the key interesting component and that these should be the focus of a performance-oriented submission to a software journal such as SoftwareX or TOMS. GMD is, of course, itself a software journal, with a focus on software applicable to geoscientific applications. We therefore don't feel that it's necessary to reorient the paper onto what TOMS or SoftwareX would contain.

Conversely since the representation of point evaluation as a finite element operation in the UFL abstraction is a core contribution of this manuscript, much of what Doug Brinkerhoff took to be unnecessary mathematical detail is actually necessary. That said, both reviewers are entirely correct that there are a number of points at which our exposition could have been significantly improved, and we will make those changes in the revised manuscript as noted below.

**2   Response to Specific Comments by Doug Brinkerhoff**

**P1** Capitalisation of Partial Differential Equations. This is has been fixed in the revised manuscript.

**P2, Para 2** Which versus that. We have gone through the manuscript and corrected this here and in a few other places.

**P2, Para 2** Differentiation meaning AD. The reviewer is correct and we have amended this accordingly.

**Sec. 2** As suggested, we have removed much of the "text book" material from chapter 2. We agree with the reviewer that it is still necessary to restate that at finite element field is a weighted sum of basis functions, because the typical GMD reader may not have the reviewer's intimate acquaintance with finite element methods.

**Sec. 3** Here we disagree with the reviewer. It is certainly true that the evaluation operator is a linear operator between finite dimensional vector spaces with specified bases, and is hence expressible as a matrix. However, the strength of the finite element method, as exploited by Firedrake, FEniCS and others, is that a high level mathematical expression of the problem to be solved can be used to automatically generate the relevant matrices, and can be algorithmically differentiated to enable the solution of nonlinear systems and optimisation problems. To simply jump to the matrix begs the question in this regard. The linear algebra approach that the reviewer advocates is not simpler if one starts from the position of vector calculus applied to functions in finite element spaces, which is the basic formalism of the finite element method. The derivations in this section are therefore not about data structures, they are about raising the level of mathematical abstraction of point cloud interpolation to that of the rest of the field. Here we note the kind description by the other reviewer of this extension of the UFL formalism as "remarkable". We have rewritten part of the introduction to make this motivation for the new formalism explicit.

Now we turn to whether our maths is actually correct. Here we think that the comments of the reviewer are dual to the correct situation. Consider a domain comprised of a cloud of points (what we call a vertex only mesh):

$$\Omega_v = \{X_i\}_{i=0}^{N-1}, \tag{1}$$

If we restrict ourselves to real scalar-valued functions for brevity of exposition (nothing important changes in the complex, or vector- or tensor-valued case) then a function defined on a vertex-only mesh, has the form:

$$f : \Omega_v \to \mathbb{R}. \tag{2}$$

$f$ associates with each vertex $X_i$ a value. The space of all such functions, $V$, is clearly $N$-dimensional. The natural finite element basis for this is:

$$\psi_i(x) = \begin{cases} 1, & x = X_i \\ 0, & \text{otherwise} \end{cases} \quad 0 <= i < N \tag{3}$$

Which is exactly the P0DG basis. A Dirac delta, in contrast, is a functional or measure. It can't be a basis function for $V$ because it doesn't lie in the space $V$. It is, however a member of $V^*$ and the corresponding dual basis in the Ciarlet sense is:

$$\psi_i^*(g) = \delta_{X_i}(g) = g(X_i) \quad 0 <= i < N \tag{4}$$

In defence of his position that the basis functions need to be Dirac deltas, the reviewer writes: "the spatial integral of a finite valued basis function that is only defined at a single point in Rd (as in Eq. 7) would be zero". This would be true if the integral were over the spatial domain $\Omega$ and the integral measure were therefore $d(> 0)$-dimensional. However, in equation 7 and elsewhere where we integrate over the point cloud, the domain of integration is $\Omega_v$ and the measure is therefore the finite sum of individual point measures making up the cloud.

Integration is a bounded linear functional, and integration of the functions defined at a single point must be linear in the single function value. It is therefore equal to the Dirac delta defined at that point. Once again, the Dirac delta is the functional rather than the function.

One possible cause of this confusion is that we have adopted the UFL convention of writing $dx$ for the volume measure of whichever domain is being integrated over, which is a point measure on a vertex only mesh. We have now disambiguated this by writing $dx_v$ to indicate the point measure when the domain of integration is $\Omega_v$.

**Sec. 4** Here we encounter the same disagreement as the previous section, so the response is essentially the same. The reviewer notes that Firedrake has a pre-existing "at" syntax, and claims that this is sufficient to abstract away the point evaluation of functions. This provides an opportune moment to explain why this is not so. The "at" method of a Function in Firedrake does indeed evaluate the function at the provided coordinates. However, it does not support the concept of a persistent set of point cloud locations nor of a set of values associated with them. This matters in the context of algorithmic differentiation and inverse problems, because in order to define the required Gateaux derivatives, we need to have a concept of known and unknown variables of a given type. In this case the type is "values associated with a particular collection of point locations". A large part of the point of these sections are the introduction of the required types (FunctionSpaces) and variables of those types (Functions). This representation of point cloud data in terms of a distinctive type is also useful for performance reasons in providing a location to cache point searches and parallel decomposition, but this is secondary to he main point here.

**Equation 17** we have prepended the conventional phrase.

**On P9, regarding pointwise evaluation of DG functions** Users frequently run problems on regular meshes and then choose evaluation points at round number coordinates that coincide with element boundaries, so this case is nowhere near as uncommon as the reviewer might suspect. The challenge to special-casing the result of interpolation in this case, as the reviewer suggests, is that roundoff error makes it exceptionally difficult, if not impossible, to detect that the point lies on a cell boundary. The most that could be done is to apply the special case to a finite width band around the edge of the cell. This would be unlikely to please users. Of course the strict answer is that point-evaluating a discontinuous finite element space is not well defined *at any point*. However this is an interpolatory crime that users frequently choose to commit, so the current behaviour of Firedrake is the compromise that best matches user expectations.

**P9, last paragraph** Yes, interpolation is a linear operator. This doesn't remove the need to represent it symbolically in a way that Dolfin-adjoint/pyadjoint can reason about.

**Eq. 19** $i$ is an index into the set of evaluation points $\{X_i\}$. We have made this explicit.

**Just after Eq. 20** The reviewer is correct that "guess" is imprecise. We have reworded in accordance with the suggestion.

**Note on nomenclature** This has been removed as suggested.

**Eq. 21** There are other ways out of the dilemma including, as the reviewer points out, using Gaussian processes. This would involve estimating the Gaussian random field f that produced the observations, and then creating a misfit term that penalises the distance to the mean of f, using the covariance operator as a metric in the norm. Nonetheless, using Gaussian processes is not yet standard practice at least in the literature on glaciological inverse problems. It is far more common to interpolate the observational data to an intermediate field in a way that does not account for the sparsity or density of observational data. Improving on this common practice is the main point of this paper. Our approach does allow for the possibility of different weights or variances for each observation point (or indeed covariances) by using a weighted norm in equation 21 and in fact we do this in the Larsen C demo at the end of the paper.

**Last line of Sec. 6** We thank the reviewer for referring us to Bouziani and Ham (2023), which one of us coauthored. That paper does not employ point evaluation and so is not directly relevant to the point being made in the last line of section 6. The line in question specifically says that automated code generation systems with adjoint capability (or differentiable programming models for the finite element method, if one chooses to adopt the terminology in Bouziani & Ham) have not previously handled point data. The reviewer objects that one could apply a generic AD tool to a finite element model. This is true, but it's a long way from an automated process and the arguments about efficiency and robustness of generic AD systems that we raised in Farrell et al. (2013) still apply. A more relevant alternative would be to hand-code the adjoint to the point interpolation and to implement a sui generis composition of that code with Firedrake's adjoint, as the authors did using Firedrake in Roberts et al. (2022). However, even that approach lacks the seamlessness, automation and expressiveness of directly incorporating a differentiable point evaluation operator in Firedrake itself.

**Between Eqs. 24 and 25** As noted above, the definition of the basis functions is correct.

**P13** We have fixed the capitalisation of L-curve.

**Sec. 7.1.1** We have rewritten this section to avoid reference to posterior consistency. The point can be made more clearly without reference to Bayesian inversion at all. The point is that if the observations are first interpolated to the computational grid and then incorporated as a misfit term penalising deviation between the interpolated data and the solution of the inverse problem, then the error will saturate as the number of observations is increased, whilst if the interpolation operator is used and the misfit term is properly defined as a sum over observation points, the error can continue to decrease with observations below the saturation point.

**Eq. 32** This was a mistake, we have corrected it in the revision.

**P18, paragraph 3** We have elaborated on this more in the text. In this example, there are only 3 parameters to infer, so much of the machinery of Bayesian inference is not necessary.

**P18, last paragraph** We have rewritten this to remove mention of posterior consistency as 7.1.1.

**Sec. 7.3.1** We disagree. Although this is not a glaciology paper as such, we feel that it is important to give some justification for why we used this simplified equation set. If we had submitted this paper to The Cryosphere then we might assume this knowledge on the part of readers.

**Sec. 7.3.2** Corrected in the revised version.

**P21, para 3** We have added a statement to the text to this effect. Ideally, we would have conducted this experiment using only the raw chip matches or displacements that are obtained from repeat-image feature tracking, which are not on a regular grid, rather than the regular grid of velocity values that are interpolated after the fact. Instead we made do with what is readily publicly available, i.e. a gridded velocity product.

**Fig. 8** This is the units for the regularisation parameter (meters).

**P23 L1** This would involve estimating the Gaussian random field f that produced the observations, and then creating a misfit term that penalises the distance to the mean of f, using the covariance operator as a metric in the norm. If the observation data are correlated, then this can be modelled using a covariance matrix in the misfit in equation 21. This would be a worthwhile exercise for a glaciology / remote sensing paper. However, the purpose of this exercise to conduct an experiment that is not possible by the conventional approach of first interpolating the observational data to the finite element mesh and then using the 2-norm misfit between the computed and interpolated velocity fields.

**P23 Para 2** We have clarified the text. Briefly, if you are willing to assume that (1) the model is correct, and (2) that the error estimates are correct in a relative but not an absolute sense, then it is possible using cross-validation to estimate the absolute scale that the error estimates should have been on.

**P23 Para 3 and 4** We have cut most of this text and revised what remains. We do not claim to have taken the most principled approach possible and we agree that a full Bayesian treatment would be superior. Instead, we argue that (1) accounting for the point-like nature of the observations, either directly in the formulation of the inference problem or by using Gaussian process regression, is a substantial improvement over interpolating and then fitting, and (2) Firedrake is now provides seamless forward and adjoint support for the point-like nature of the observations, which is relatively uncommon among finite element modelling packages. We do not need to take the most principled approach imaginable in order to improve upon existing practice in the glaciological community or in other disciplines. Point data assimilation enables substantial improvements to current practice.

**Sec. 8** We think that the reviewer's core question about this section reflects the fact that we were insufficiently clear that the point data abstraction and implementation are core to the paper. That said, we have revised this section to make it clearer that it is about further work which is facilitated by a point data abstraction. The section has also been updated to account for development which has occurred in the extended period that the paper has been in review.

**3 Response to Specific Comments by Umberto Villa**

1. The use of unstructured point data in inverse problems and data assimilation is well established. For example, the hIPPYlib software [1,2,3], an inverse problem Python library based on FEniCS (of which I am a developer), already implements point-wise observations while automating the computations of first- and second-order derivatives by use of FEniCS symbolic differentiation of weak forms. Furthermore, the three examples presented in the paper are related to some applications of hIPPYlib, including the inversion for a diffusion coefficient in the Poisson equation [1], poroelasticity [4], and ice sheet flow [5].

We have cited hIPPYlib and explained the difference between that work and the approach presented here. We are not the first to assimilate point data and we do not claim to be – oceanographers have assimilated data from drifting buoys into circulation models for years. The key distinction between this and previous work such as hIPPYlib is that the point data is included as fully first class objects in the symbolic layer of the finite element system. This enables data assimilation, which hIPPYlib also supports, but also enables point data to be used almost anywhere in a calculation where any other finite element function could be used. For example, this enables the interpolation to point data to be put in the integrand, as shown in equation 13. The beauty of introducing a fully composable first class operator is that users will certainly find uses for this functionality that we have not thought of.

2. In the numerical results, the choice of the interpolation methods to map the discrete observation to the grid is naïve. A better approach would have been to compare with Gaussian process interpolation (and its 2D/3D counterparts). Gaussian processes do account for uncertainty in the measurement and provide a spatially varying estimate of the variance of the interpolator, thus allowing for more consistent handling of uncertainty measurement and of the fact that the interpolation is less reliable far away from observation points.

The goal of this experiment is to compare the approach the we propose with what is commonly done in the literature. Many publications have used naive interpolation methods to map discrete observations to a finite element mesh without acknowledging the impact that this arbitrary algorithmic choice can make. We agree that using Gaussian process regression would be superior to naive interpolation, but relatively few publications (at least in the glaciology literature) use it.

3. The authors claim that the numerical results demonstrate Bayesian consistency of the proposed interpolation method. Why this claim is true, the numerical results do not support such claim as the authors do not solve a Bayesian inverse problem

The reviewer correctly points out several instances where we overstated the statistical rigour of our experiment. We have walked back some of these claims. The purpose of the demonstrations here is to show that the new point data assimilation capabilities of Firedrake work as advertised and that this feature is fully integrated into the rest of the symbolic capabilities of the library. We claim that this new feature is useful whether one is interested in solving problems from the viewpoint of both Bayesian statistics or deterministic inverse problems.

1. The authors only compute the MAP point, while the solution of the Bayesian inverse problem requires the ability to characterize (either by sampling or using variational inference the posterior distribution.

We agree, we overstated our case here and have altered the text accordingly.

2. The authors use Tikhonov regularization on the spatial gradient of the un- known parameter as the logarithmic of the prior distribution. However, since the inverse of the Laplace operator in two or three spatial dimensions is not a trace-class operator, the Bayesian inverse problem is not well-posed with that choice of prior [6].

This was an oversight on our part and we thank the reviewer for pointing this out. Using the square norm of the gradient is an inverse crime and we have updated the text to reflect this. Nonetheless, the essential point of this exercise is to demonstrate the new point data assimilation capabilities added to Firedrake and that the resulting estimates converge in the limit of a large number of observations. Using the square norm of the second derivative is better practice but does not substantially alter the essential point, so we have kept the simulations as-is.

3.        The L-curve, Morozov, or cross-validation criterion are commonly used in the deterministic formulation of inverse problems. In the Bayesian formulation, hyper-priors are often used to tune the variance and correlation length of the prior distribution [7].

We have removed discussion of a Bayesian inverse problem.

4.        The definition of Bayesian consistency is not interpreted correctly. Below formula (31) the authors claim that as the number of observation points goes to infinity then $q_{\text{est}}$ should converge toward $q_{\text{true}}$. This is not what Bayesian consistency is. In fact, there are several cases in which it is impossible to recover the true parameter exactly no matter how many observation points are used, simply because there is an intrinsic loss of information in the continuous map from the parameter to the state variable.

Please see our answer to reviewer Brinkhoff about Sec. 7.1.1.

5.        It is not clear what is the solution approach in the second example. Are the authors employing a Kalman filter or are they solving for a MAP point and then invoking the Laplace approximation to the posterior to estimate the posterior PDF?

We have expanded on the explanation in the text. We are not using a Kalman filter or the Laplace approximation. Briefly, there are only three parameters to infer and 18 observations, so computing a MAP estimator with the improper uniform prior on the transmissivities is the same as computing the maximum likelihood estimator. We generated 30 independent realisations of the observation set, computed a MAP/MLE from each one, and then fit a normal distribution to the resulting 30 estimates for the transmissivities.

6.        Given that the main contribution of the paper is the implementation in UFL/Firedrake of a point-wise observation operator, it would be good to numerically demonstrate the scalability of the implementation with respect to the number of observation points and the number of MPI processes for both 2D and 3D geometries.

The reviewer suggests that scaling experiments are desirable. We suggest that this is not actually valuable in the context of the functionality presented here. The reason is that point evaluation at static points typically accounts for a vanishing proportion of runtime. The circumstances where the performance of point evaluation are likely to become a first order concern are where there are a very large number of particles and these move (hence necessitating frequent updates of the containing cell and local coordinates). This functionality is not presented here and is listed in the future work section. A future paper focussed on, for example, statistics of moving particles would be the appropriate juncture to study the performance of the system.

7.   1. The reviewer raised concerns about consistency of the notation for the various functionals. We have standardised the notation to use $J^{\text{point}}$ and $J^{\text{field}}$

    2.        Section 2 is a mix between the very basic theory of finite elements and more advanced concepts like dual basis. However, the section lacks of mathematical rigor in the notation and there is an imbalance of content. In particular, the description of the foundation of finite elements is too long and well-known (e.g. Fig 1 could be removed), while the most advanced most relevevant to this work (the dual basis and the Charlet triplet) lack of details.

    This is a well-made point. We have pared back this section significantly.

    3. We have tightened up the formality of the mathematical exposition throughout. We would like to point that the semicolons are not typos but are an indication that the operator in question is linear in the arguments after the semicolon. This notation was introduced at the end of chapter 3 and is an extension of the notation for forms used in, for example, Alnæs et al. (2014). The use of tilde to indicate quantities in local coordinates has been made consistent.

       The presentation of the paper is very informal (e.g. the use of "our", to refer to "our" finite element space, "our" finite element function, "our" inverse problem) and there are several typos in the mathematical formulas (e.g. there appear to be an extra semicolon in Eq (11) and (12); or inconsistency between $\tilde{x}$ in Eq (13) and the text below where $\hat{x}$ is used.

**References**

Alnæs, M. S., Logg, A., Ølgaard, K. B., Rognes, M. E., and Wells, G. N.: Unified form language: A domain-specific language for weak formulations of partial differential equations, ACM Transactions on Mathematical Software, 40, 9:1–9:37, https://doi.org/10.1145/2566630, 2014.

Bouziani, N. and Ham, D. A.: Physics-driven machine learning models coupling PyTorch and Firedrake, in: ICLR 2023 Workshop on Physics for Machine Learning, https://doi.org/https://doi.org/10.48550/arXiv.2303.06871, 2023.

Farrell, P. E., Ham, D. A., Funke, S. W., and Rognes, M. E.: Automated Derivation of the Adjoint of High-Level Transient Finite Element Programs, SIAM Journal on Scientific Computing, 35, C369–C393, https://doi.org/10.1137/120873558, publisher: Society for Industrial and Applied Mathematics, 2013.

Roberts, K. J., Olender, A., Franceschini, L., Kirby, R. C., Gioria, R. S., and Carmo, B. S.: spyro: a Firedrake-based wave propagation and full-waveform-inversion finite-element solver, Geoscientific Model Development, 15, 8639–8667, https://doi.org/10.5194/gmd-15-8639-2022, 2022.